# A fructose/H$^+$ symporter controlled by a LacI-type regulator promotes survival of pandemic *Vibrio cholerae* in seawater

Yutao Liu[1,2,3,6], Bin Liu[1,2,3,6], Tingting Xu [1,2,3,4,6], Qian Wang[1,2,3], Wendi Li[1,2,3], Jialin Wu[1,2,3], Xiaoyu Zheng[1,2,3], Bin Liu[1,2,3], Ruiying Liu[1,2,3], Xingmei Liu[1,2,3], Xi Guo[1,2,3], Lu Feng [1,2,3✉] & Lei Wang [1,2,3,5✉]

The bacterium *Vibrio cholerae* can colonize the human intestine and cause cholera, but spends much of its life cycle in seawater. The pathogen must adapt to substantial environmental changes when moving between seawater and the human intestine, including different availability of carbon sources such as fructose. Here, we use in vitro experiments as well as mouse intestinal colonization assays to study the mechanisms used by pandemic *V. cholerae* to adapt to these environmental changes. We show that a LacI-type regulator (FruI) and a fructose/H$^+$ symporter (FruT) are important for fructose uptake at low fructose concentrations, as those found in seawater. FruT is downregulated by FruI, which is upregulated when O$_2$ concentrations are low (as in the intestine) by ArcAB, a two-component system known to respond to changes in oxygen levels. As a result, the bacteria predominantly use FruT for fructose uptake under seawater conditions (low fructose, high O$_2$), and use a known fructose phosphotransferase system (PTS, Fpr) for fructose uptake under conditions found in the intestine. PTS activity leads to reduced levels of intracellular cAMP, which in turn upregulate virulence genes. Our results indicate that the FruT/FruI system may be important for survival of pandemic *V. cholerae* in seawater.

[1] The Key Laboratory of Molecular Microbiology and Technology, Ministry of Education, Tianjin, P. R. China. [2] TEDA Institute of Biological Sciences and Biotechnology, Nankai University, TEDA, Tianjin, P. R. China. [3] Tianjin Key Laboratory of Microbial Functional Genomics, Tianjin, P.R. China. [4] Shenzhen Institute of Respiratory Diseases, Second Clinical Medical College (Shenzhen People's Hospital), Jinan University, Shenzhen, P. R. China. [5] State Key Laboratory of Medicinal Chemical Biology, Nankai University, Tianjin, P. R. China. [6] These authors contributed equally: Yutao Liu, Bin Liu, Tingting Xu. ✉email: fenglu63@nankai.edu.cn; wanglei@nankai.edu.cn

The gram-negative facultative anaerobic bacterium *Vibrio cholerae* is the causative agent of cholera, a severe water-borne diarrheal disease that has caused seven pandemics since 1817. The current, seventh pandemic started in 1961 and continues to be a significant cause of death globally, with 3–5 million new infections and more than 100,000 deaths every year[1]. Among pandemic strains, only those from the sixth and seventh cholera pandemics are available for modern scientific study, and phylogenetic analyses suggested that pandemic strains originated from nonpandemic strains[2,3]. *V. cholerae* has a complex life cycle involving transitions between the aquatic environment, such as surface seawater, and the human small intestine[4]. *V. cholerae* exhibits the ability to survive in the aquatic environment year-around[5,6]. To survive in the environment, *V. cholerae* has to respond to changing conditions, including temperature shifts, osmotic stress, and nutrient limitation. The bioavailability of carbon sources is limited in seawater[7]. Several strategies enabling *V. cholerae* to acquire or store carbon sources in this environment have been reported, such as utilizing chitin in the exoskeleton of marine crustaceans and using accumulated intracellular glycogen stores[6]. In addition, extracellular DNA and dissolved organic matter of seawater may also support the growth of *V. cholerae*[8,9]. However, the mechanisms that *V. cholerae* utilizes to efficiently uptake and metabolize carbon sources of seawater are still not fully understood.

Once ingested by humans, *V. cholerae* passes through the stomach, grows in the small intestine, transits through the mucus, colonizes the surface of the intestinal epithelial cells, and causes severe diarrhea[10]. *V. cholerae* undergoes a significant shift in environmental conditions as it transitions from the aquatic environment into and within the human host. Adaptation to these changes involves changes in gene expression including the precise induction of virulence genes in the small intestine[11]. Like other enterobacteria[12,13], *V. cholerae* mainly utilizes sugars present in the human intestine as the major carbon and energy sources[14,15].

In bacteria, there are three types of active sugar transport systems: phosphoenolpyruvate: carbohydrate phosphotrasferase system (PTS), major facilitator superfamily (MFS) transporters, and ATP-binding cassette transporters. There are 13 distinct PTS transporters in *V. cholerae*, including a PTS specific for fructose[16,17]. Sugar transport via PTS lowers the activity of adenylate cyclase that is responsible for the synthesis of cyclic adenosine monophosphate (cAMP)[18]. cAMP receptor protein (CRP) binding with cAMP negatively regulates the expression of several virulence genes in *V. cholerae*[19–21]. MFS transporters catalyze sugar translocations across the membrane against its concentration gradient by using the energy released from the energetically downhill movement of a cation[22]. LacI-type regulators play important roles in bacterial carbon utilization[23]. They can coordinate available sugars with the expression of corresponding transport and catabolic genes, benefiting bacteria by conserving energy for mRNA and protein production[24,25]. There is a huge difference in the availability and concentration of sugars between seawater and human intestine[12,26,27], and it remains to be understood whether LacI-type regulators play a role in regulating sugar uptake and utilization of pandemic *V. cholerae* for adapting to changes between these two environments.

Previous studies on pandemic *V. cholerae* strains have mostly focused on the identification of virulence factors expressed by the bacteria in the human intestine; these studies have shown that the generation and evolution of pandemic strains is associated with the acquisition of virulence elements, such as the *ctx* phage encoding cholera toxin, *Vibrio* pathogenicity island (VPI), and *Vibrio* 7th Pandemic islands I and II (VSP-I and II)[2,28]. Pandemic *V. cholerae* strains also acquired several abilities that benefit their

survival and persistence in the environment, such as formation of biofilms[6]. It is highly likely that acquisition of factors to efficiently uptake and utilize sugars is essential for the survival of pandemic *V. cholerae* in the aquatic environment where carbon source is limited; however, this was often overlooked in previous research studies.

Here we aimed to identify the carbon uptake transporter that might confer an advantage to pandemic *V. cholerae* strains as well as the mechanisms by which carbon transport is regulated during the bacteria life cycle. We first performed a large-scale comparative genomics analysis on all 762 available pandemic *V. cholerae* genomes and 433 available nonpandemic *V. cholerae* genomes. We found that *fruT* encoding a fructose/$H^+$ symporter and *fruI* encoding a LacI-type regulator that negatively regulates *fruT* expression are present in all pandemic strains, while are only present in 27.3% of nonpandemic strains. Fructose, whose concentration is in the range of 0.12–0.55 μM, is a major sugar present in seawater[26]. Fructose uptake assays revealed that FruT is responsible for uptaking low-concentration fructose. We showed that in seawater, the expression of *fruT* is turned on as the repression of *fruT* by FruI is released in response to high $O_2$ concentration, and deletion of *fruT* made pandemic *V. cholerae* exhibit a substantial survival disadvantage in seawater. Animal experiments showed that deletion of *fruI* significantly inhibited colonizing ability of *V. cholerae* in the small intestine of mice. Through a series of molecular biology experiments, we showed that in the small intestine, FruI directly downregulates *fruT* expression by sensing the low $O_2$ signal. This promotes the uptake of high-concentration fructose via PTS in the small intestine, and enhanced PTS activity leads to a decrease in the intracellular cAMP level, and upregulation of virulence gene expression. In summary, our work revealed that acquisition of this fructose uptake system not only promotes bacterial survival in seawater, but also maintains bacterial pathogenicity in animal intestine.

## Results

**FruI is a LacI-type regulator involved in fructose utilization.** There are 11 genes encoding LacI-type regulators (VCA0673, VC0289, VC0909, VC1286, VCA0132, VCA0519, VC2677, VC2337, VC1721, VC1557, and VCA0654) in the genome of *V. cholerae* strain N16961, a representative strain of the seventh cholera pandemic[29]. To reveal whether some of these 11 LacI-type regulator genes are more associated with pandemic *V. cholerae* than nonpandemic *V. cholerae*, a comparative genomics analysis was performed using all 762 publicly available genomes of pandemic strains and 433 publicly available genomes of nonpandemic strains. The comparative genomics analysis showed that these 11 genes are present in all pandemic strains (Supplementary Data 1). They are also present in all or most nonpandemic strains, except for VCA0673 that is present in 27.3% of nonpandemic strains (Supplementary Data 1). It suggests that VCA0673 is considerably associated with pandemic strains and may play a role in the pathogenicity and/or environmental survival of pandemic strains. However, the function of VCA0673 remains unclear.

BlastP analysis showed that VCA0673 shares sequence similarity to LacI-type regulators FruR, TreR, MalR, and RbsR, which regulate the transport and/or utilization of fructose, trehalose, maltose, and ribose, respectively. To assess whether VCA0673 is indeed involved in the transport and/or utilization of these sugars, we constructed a deletion mutant strain Δ*vca0673* in strain El2382, a seventh-pandemic strain stored in our lab whose genome has been sequenced[30]. Then we assessed how deletion of this gene affected growth of *V. cholerae* in medium with each of

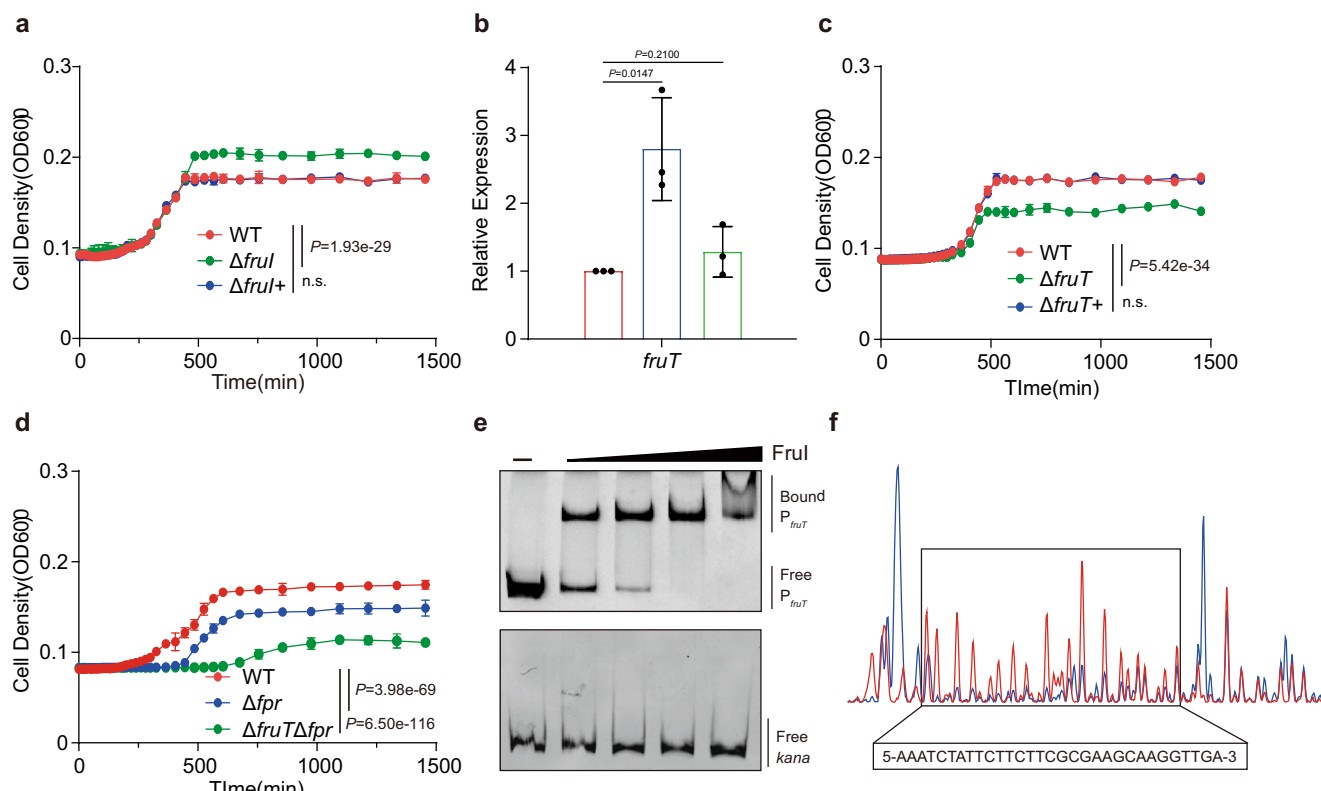

**Fig. 1 FruI represses the expression of the putative fructose transporter FruT. a** Growth curves of WT, Δ*fruI*, and Δ*fruI*+ in M9 medium containing 30 mM fructose as the only carbon source. Data represent the mean ± SD ($n = 3$). **b** qRT-PCR expression level of *fruT* in WT, Δ*fruI*, and Δ*fruI*+ in M9 medium containing 30 mM fructose as the only carbon source. Data represent the mean ± SD ($n = 3$). **c** Growth curves of WT, Δ*fruT*, and Δ*fruT*+ in M9 medium containing 30 mM fructose as the only carbon source. Data represent the mean ± SD ($n = 3$). **d** Growth curves of WT, Δ*fpr*, and Δ*fruT*Δ*fpr* in M9 medium containing 30 mM fructose as the only carbon source. Data represent the mean ± SD ($n = 3$). **e** EMSA of the specific binding of purified FruI protein to the promoter region of *fruT* and *kana* (negative control). **f** FruI binds to a motif in the *fruT* promoter region. The protected region shows a significantly reduced peak intensities (blue) pattern than seen in compared with those of the control (red). The identified FruI-binding motif is shown in a box at the bottom of the figure. Images are representative of three independent experiments (**e**). Significance is determined by two-tailed unpaired Student's *t* test (**b**) or two-way ANOVA (**a**, **c**, **d**) and indicated as the *P* value; n.s. means no significant difference. Source data are included in Source Data file.

those four sugars as the only carbon source, by growing the wild-type (WT) strain and Δ*vca0673* in M9 medium supplemented with 10 or 30 mM fructose, trehalose, maltose, or ribose, respectively. There is no significant difference in bacterial growth curves between Δ*vca0673* and WT in M9 medium supplemented with trehalose, maltose, or ribose, respectively, indicating VCA0673 is not involved in the regulation for the transport and utilization of these sugars (Supplementary Fig. 1a–h). However, in M9 medium supplemented with fructose, the bacterial growth curves showed that Δ*vca0673* underwent a longer exponential phase than WT, and reached a higher final cell density than WT (Fig. 1a and Supplementary Fig. 1i). This effect was reversed by complementation of the Δ*vca0673* (Fig. 1a and Supplementary Fig. 1i). This result suggests that Δ*vca0673* exhibits better ability to utilize fructose than WT. Thus, the LacI-type regulator encoded by *vca0673* may be involved in the regulation for fructose transport and/or utilization in *V. cholerae* and, consequently, the gene was named *fruI* and the Δ*vca0673* was renamed Δ*fruI*.

**FruI represses the expression of the putative fructose transporter FruT by directly binding to its promoter.** To investigate the molecular mechanisms through which FruI regulates fructose transport and/or utilization, we compared the global gene expression of Δ*fruI* and WT at the stationary phase in M9

medium supplemented with 30 mM fructose as the only carbon source. We identified 456 genes that were significantly differentially expressed between Δ*fruI* and WT (Supplementary Data 2). Among these 456 genes, *vca0669* is the most upregulated gene (394-fold) in Δ*fruI* compared to that of WT. *vca0669* is located 1.3 kb upstream of *fruI* (Supplementary Fig. 1k), and both of them are within a genomic island (GI-7) revealed by comparative genomics analysis on *V. cholerae*[31]. qRT-PCR assays confirmed that the expression of *vca0669* exhibited a 2.8- and 17.4-fold increase in Δ*fruI* compared to WT at the stationary phase in M9 medium supplemented with 10 or 30 mM fructose (Fig. 1b and Supplementary Fig. 1l). Complementation of Δ*fruI* with *fruI* restored *vca0669* expression to the WT level (Fig. 1b). VCA0669 encodes a potential MFS sugar transporter[29]. Bacterial growth curves showed that deletion of *vca0669* did not influence bacterial growth in M9 medium supplemented with 10 or 30 mM glucose, trehalose, maltose, and ribose, respectively (Supplementary Fig. 1a–h). However, in M9 medium supplemented with 10 or 30 mM fructose, Δ*vca0669* underwent a shorter exponential phase than WT, and the final cell density of Δ*vca0669* was also lower than that of WT (Fig. 1c and Supplementary Fig. 1j), indicating the deletion of *vca0669* impairs bacterial ability to utilize fructose partially. This effect was reversed by complementation of Δ*vca0669* (Fig. 1c and Supplementary Fig. 1j). These data suggest that FruI represses the expression of *vca0669* and that *vca0669*

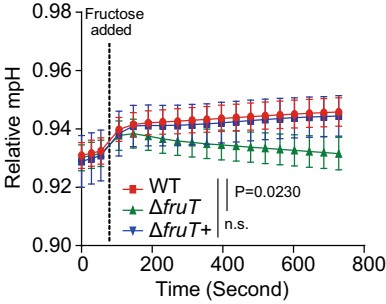

**Fig. 2 Fructose symport activity of FruT.** Effect on the extracellular pH elicited measured before and after the addition of fructose to a final concentration of 30 mM to unbuffered aqueous cell suspensions of WT, Δ*fruT*, and Δ*fruT*+. The extracellular pH was measured by Seahorse XF analyzer. The dotted lines indicate the times of fructose addition. Data are shown as the mean ± SD ($n = 3$). Significance is determined by two-way ANOVA and indicated as the *P* value; n.s. means no significant difference. Source data are included in Source Data file.

may encode a fructose transporter. Consequently, *vca0669* was named *fruT* and the Δ*vca0669* and complemented strains were named Δ*fruT* and Δ*fruT*+, respectively.

Since the Δ*fruT* could still grow in M9 medium with fructose, even though it entered into stationary phase earlier than the WT, we evaluated if fructose PTS plays a role during the exponential phase growth. *fpr* encodes a multidomain protein Fpr included in fructose PTS of *V. cholerae*, which consists of an enzyme IIA (EIIA)-like domain and a HPr domain[17]. Enzyme I (EI) autophosphorylates in the presence of phosphoenolpyruvate and then subsequently transfers this phosphate to FPr[17]. Bacterial growth curves in M9 medium supplemented with 10 or 30 mM fructose showed that both Δ*fpr* and ΔEI exhibited a lower growth rate than WT in the exponential phase, and reached a lower final cell density than WT (Supplementary Fig. 1m, n). In addition, Δ*fruT*Δ*fpr* barely grew in M9 medium supplemented with 10 or 30 mM fructose, which reached a 1.55- to 1.83-fold lower final cell density than that of WT, Δ*fpr*, and Δ*fruT* (Fig. 1d and Supplementary Fig. 1o), indicating that both FruT and PTS are required for pandemic *V. cholerae* growth with fructose as the only carbon source.

To evaluate whether FruI directly regulates *fruT* expression, we performed electrophoretic mobility shift assays (EMSAs) using purified 6×His-tagged FruI and the potential promoter region of *fruT* corresponding to 200 bp upstream of *fruT*. The result showed that FruI specifically binds to the *fruT* promoter (Fig. 1e). A dye-based DNase I footprinting assay revealed a specific FruI-bound sequence containing an 31–base pair motif (5-AAATC-TATTCTTCTTCGCGAAGCAAGGTTGA-3; −106 to −74 from the *fruT* translational start site) in the *fruT* promoter region (Fig. 1f). These data indicate that FruI in *V. cholerae* directly downregulates the expression of *fruT*.

**FruT is a H⁺ symporter.** To investigate the specific function of FruT, we performed protein domain structure analysis using SMART, which indicates that FruT is a hypothetical H⁺ symporter. We also found that FruT shares 32.6% similarity with MstA, a H⁺ symporter identified in *Aspergillus niger*, which is able to transporter the sugar simultaneously with the movement of protons[32]. To corroborate this finding, symport assays were performed to investigate whether fructose transport mediated by FruT is associated with H⁺ movement. Changes in the extracellular pH of unbuffered cell suspensions of WT, Δ*fruT*, and Δ*fruT*+ upon the addition of fructose were monitored. Significant extracellular alkalinization reflecting a reduction in the

extracellular H⁺ concentration was observed in the media of WT and Δ*fruT*+ but not in the medium of Δ*fruT* (Fig. 2). These data suggest that FruT transports fructose by a H⁺ symport mechanism.

**FruI contributes to the virulence of *V. cholerae* by repressing the expression of *fruT* and decreasing the intracellular cAMP level.** The above findings indicated that FruT likely encodes a fructose/H⁺ symporter that is required for *V. cholerae* to grow in fructose in vitro; we also found that *fruT* expression is negatively regulated by FruI, whose deletion enhances *V. cholerae* growth in fructose in vitro. Next, we investigated whether FruI and FruT are required for *V. cholerae* colonization in small intestine by performing competitive infection assays in mice. The competitive index (CI) value of Δ*fruI* versus WT was 0.45 in the small intestine of mice (Fig. 3a), while the CI value of complemented strain (Δ*fruI*+) versus WT was 1.19 (Fig. 3d). This indicates that compared with the WT, Δ*fruI* exhibits a significantly defective colonizing ability in the small intestine and that FruI contributes to intestinal colonization of *V. cholerae*. The CI value of Δ*fruT* versus WT was 3.18 in the small intestine of mice (Fig. 3b), while the CI value of Δ*fruT*+ versus WT was 0.86 (Fig. 3d), indicating that Δ*fruT* exhibited a significant colonization advantage over WT. Furthermore, the competitive infection assays showed that the CI value of the double mutant Δ*fruI*Δ*fruT* versus WT was 3.26 in the small intestine of mice, which is similar to the CI value of Δ*fruI* versus WT (Fig. 3c), and Δ*fruI*Δ*fruT* competed evenly with Δ*fruT* in the small intestine of mice (CI = 1.12) (Fig. 3e). These data indicate that in the absence of the *fruT*, deletion of *fruI* does not have an effect on *V. cholerae* colonizing ability. In addition, qRT-PCR assays showed that *fruT* expression exhibited a significant increase in Δ*fruI* compared to WT in the small intestine of mice, indicating FruI represses the expression of *fruT* in the small intestine (Fig. 3f). These data suggest that FruI contributes to intestinal colonization of *V. cholerae* by down-regulating *fruT* expression.

To understand the mechanism underlying the colonization advantage of Δ*fruT*, we analyzed the expression of the virulence genes *toxR*, *tcpP*, and *toxT*, which is required for intestinal colonization of *V. cholerae*, by qRT-PCR in Δ*fruT* compared to WT in M9 medium supplemented with 20 mM fructose, which mimics the concentration of fructose in the small intestine environment[33], and in the small intestine of mice. We found that the expression of the virulence genes was significantly higher in Δ*fruT* compared to the WT strain in vitro and in vivo (Fig. 3g, h). In consistent, western blotting assays showed that the production of cholera toxin, a major virulence factor of *V. cholerae*, was significantly higher in Δ*fruT* compared to WT in M9 medium supplemented with 20 mM fructose (Fig. 3i). These results suggest that in the absence of *fruT*, the increased expression of virulence factors confers a colonization advantage to Δ*fruT*. Conversely, the increased expression of *fruT* in Δ*fruI* could explain the defective colonizing ability of this strain (Fig. 3a), as the expression of the virulence genes and the production of cholera toxin analyzed above was significantly lower in Δ*fruI* compared to WT in M9 medium supplemented with 20 mM fructose and in the small intestine of mice (Fig. 3g, h). qRT-PCR assays and western blotting assays also showed that the expression of virulence genes or production of cholera toxin in Δ*fruT* and Δ*fruI* exhibited no significant difference compared to that of WT in M9 medium supplemented with 20 mM glucose (Supplementary Fig. 2a, b), indicating that FruT and FruI influence expression of virulence factors via modulating bacterial fructose uptake.

The findings above suggested that the fructose transporter FruT was not required for *V. cholerae* growth in the intestine; in

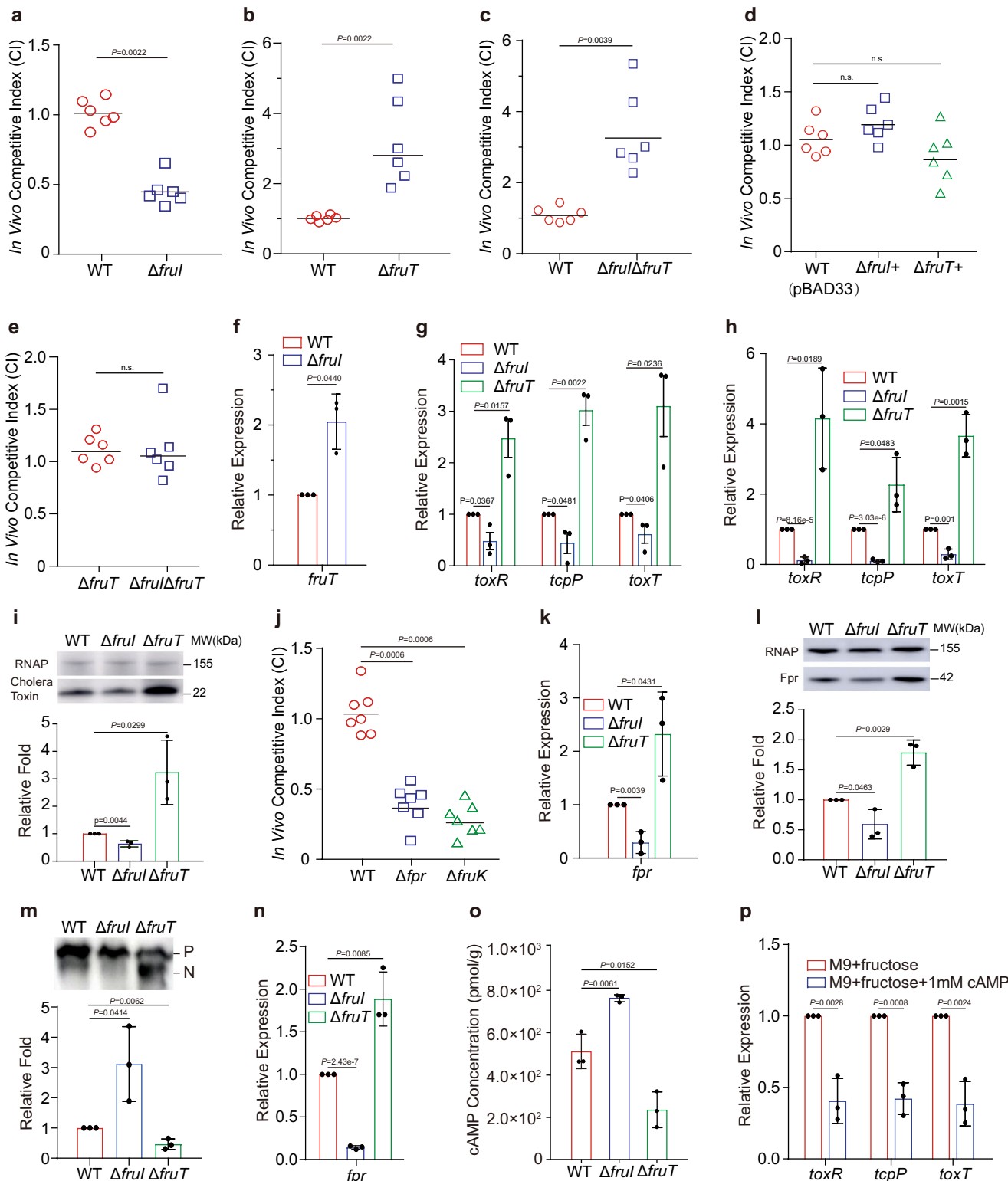

fact, ΔfruT showed a colonization advantage compared to the WT strain, raising the question of whether *V. cholerae* actually uses fructose via PTS when growing in the small intestine. To this end, we performed competitive infection assays in mice between WT and Δfpr and between WT and ΔfruK; *fruK* encodes fructose-1-phosphate kinase, which converts fructose-1-phosphate to fructose-1,6-bisphosphate in one of the key steps of the fructose catabolism[34]. The results showed that the CI values of Δfpr and ΔfruK versus WT were 0.39 and 0.28, respectively (Fig. 3j), in the

small intestine of mice. These results indicate that Δfpr and ΔfruK are significantly defective in the ability to colonize the small intestine of mice and that fructose catabolism and fructose transport via the PTS are essential for intestinal colonization by *V. cholerae*.

The above findings suggested that *V. cholerae* utilizes PTS but not FruT to uptake fructose in the small intestine of mice, and the expression of *fruT* is downregulated by FruI in the small intestine of mice. Then, we investigated whether the repression of *fruT*

**Fig. 3 FruI contributes to the virulence of *V. cholerae* by repressing the expression of *fruT* and decreasing the intracellular cAMP level. a–d** Competition assay comparing the colonizing ability of WT, ΔfruI (**a**), ΔfruT (**b**), ΔfruIΔfruT (**c**), ΔfruI+ (**d**), and ΔfruT+ (**d**) in infant mouse intestine (n = 6 mice per group). CI is defined as the output ratio of mutant strains to WT *lacZ*- divided by the input ratio of mutant strains to WT *lacZ*-, or output ratio of complemented strains to WT *lacZ*- with pBAD33 divided by the input ratio of complemented strains to WT *lacZ*- with pBAD33. Each symbol represents the CI in an individual mouse; horizontal bars indicate the median. **e** Competition assay comparing the colonizing ability of ΔfruIΔfruT and ΔfruT in infant mouse intestine (n = 6 mice per group). CI is defined as the output ratio of ΔfruIΔfruT or ΔfruT to ΔfruT *lacZ*- divided by the input ratio of ΔfruIΔfruT or ΔfruT to ΔfruT *lacZ*-. Each symbol represents the CI in an individual mouse; horizontal bars indicate the median. **f** qRT-PCR expression level of *fruT* in WT and ΔfruI in mouse small intestine. Data represent the mean ± SD (n = 3). **g, h** qRT-PCR expression level of virulence genes in WT, ΔfruT, and ΔfruI in M9 medium containing 20 mM fructose as the only carbon source (**g**) and in mouse small intestine (**h**). Data represent the mean ± SD (n = 3). **i** Representative western blotting image and quantitative analysis of cholera toxin in WT, ΔfruT, and ΔfruI in M9 medium containing 20 mM fructose as the only carbon source. RNA polymerase (RNAP) was used as a loading control. Data represent the mean ± SD (n = 3). **j** Competition assay comparing the colonizing ability of WT, Δfpr, and ΔfruK in infant mouse intestine (n = 7 mice per group). CI is defined as the output ratio of these strains to WT *lacZ*- divided by the input ratio of these strains to WT *lacZ*-. Each symbol represents the CI in an individual mouse; horizontal bars indicate the median. **k** qRT-PCR expression level of *fpr* in WT, ΔfruI, and ΔfruT in M9 medium containing 20 mM fructose as the only carbon source. Data represent the mean ± SD (n = 3). **l** Representative western blotting image and quantitative analysis of Fpr in WT, ΔfruT, and ΔfruI in M9 medium containing 20 mM fructose as the only carbon source. RNA polymerase (RNAP) was used as a loading control. Data represent the mean ± SD (n = 3). **m** Phosphorylation status analysis of Fpr in WT, ΔfruI, and ΔfruT in M9 medium containing 20 mM fructose as the only carbon source. P phosphorylated protein, N non-phosphorylated protein. Data represent the mean ± SD (n = 3). **n** qRT-PCR expression level of *fpr* in WT, ΔfruI, and ΔfruT in mouse small intestine. Data represent the mean ± SD (n = 3). **o** cAMP concentrations in WT, ΔfruI, and ΔfruT in M9 medium containing 20 mM fructose as the only carbon source. Data represent the mean ± SD (n = 3). **p** qRT-PCR expression levels of virulence genes in ΔfruT in M9 medium containing 20 mM fructose with or without 1 mM cAMP. Data represent the mean ± SD (n = 3). Significance is determined by two-sided Mann–Whitney U test (**a–e**, **j**) or two-tailed unpaired Student's t test (**f–l**, **k–p**) and indicated as the P value; n.s. means no significant difference. Source data are included in Source Data file.

leads to the enhanced fructose PTS expression and activity in vitro and in vivo. qRT-PCR and western blotting assays showed that *fpr* expression and Fpr protein level exhibited a significant increase in ΔfruT but exhibited a significant decrease in ΔfruI compared to WT in M9 medium supplemented with 20 mM fructose (Fig. 3k, l), while *fpr* expression and Fpr protein level in ΔfruT or ΔfruI exhibited no difference compared to that of WT in M9 medium supplemented with 20 mM glucose (Supplementary Fig. 2c, d). The decrease of phosphorylation level of EIIA is directly correlated with the increase of the sugar uptake mediated by PTS[35]. Western blotting showed that the phosphorylation level of Fpr, which consists of a EIIA-like domain, in ΔfruT and ΔfruI growing in M9 medium supplemented with 20 mM fructose was significantly decreased or increased compared to that of WT (Fig. 3m), while Fpr in ΔfruT, ΔfruI, and WT exhibited no phosphorylation in M9 medium supplemented with 20 mM glucose (Supplementary Fig. 2e). In addition, qRT-PCR assays showed that the expression of *fpr* exhibited a significant increase and a significant decrease in ΔfruT and ΔfruI, respectively, compared to WT in the small intestine of mice (Fig. 3n). These data indicate that deletion of *fruI*, resulting in *fruT* overexpression, inhibits the expression and utilization of the fructose PTS, while deletion of *fruT* promotes the expression and utilization of the fructose PTS.

Since CRP that responds to intracellular cAMP level is a key regulator of metabolic and virulence gene expression in bacteria in response to environmental signals, and that sugar transport via PTS is known to lower the activity of the adenylate cyclase responsible for the synthesis of cAMP[18,36], we investigated whether *fruI* and *fruT* influence the intracellular cAMP level. We found that when growing in M9 medium supplemented with 20 mM fructose, which mimics the concentration of fructose in the small intestine environment, ΔfruI exhibited significantly higher intracellular cAMP level than WT (Fig. 3o), while ΔfruT exhibited significantly lower intracellular cAMP level than WT (Fig. 3o). In contrast, the intracellular cAMP level of ΔfruI and ΔfruT exhibited no difference compared to that of WT in M9 medium supplemented with 20 mM glucose (Supplementary Fig. 2f). Consistent with cAMP-CRP negatively regulating the expression of virulence genes in *V. cholerae*[36], the above finding showed that the expressions of *toxR*, *tcpP*, and *toxT* were upregulated or

downregulated in ΔfruT and ΔfruI, respectively, compared to WT, both in the small intestine of mice and in M9 medium supplemented with 20 mM fructose (Fig. 3g, h). We also showed that the upregulation of virulence gene expression in ΔfruT was inhibited by addition of 1 mM cAMP to M9 medium supplemented with 20 mM fructose (Fig. 3p). These results suggest that deletion of *fruI* and *fruT*, which inhibits or promotes the expression and utilization of the fructose PTS, increases or decreases the intracellular cAMP level, which in turn downregulates or upregulates the expression of virulence genes.

To confirm that FruI and FruT influence the virulence of *V. cholerae* by regulating the intracellular cAMP level in the small intestine of mice, we performed competitive infection assays in mice between a mutant in the adenylate cyclase gene *cyaA* (ΔcyaA) and the double mutants ΔfruIΔcyaA and ΔfruTΔcyaA, respectively. We found that ΔfruIΔcyaA and ΔfruTΔcyaA competed evenly with ΔcyaA in the small intestine of mice (CI = 1.25 and 1.09, respectively) (Supplementary Fig. 2g, h), suggesting that in the absence of the adenylate cyclase, deletion of *fruI* or *fruT* does not have an effect on *V. cholerae* colonizing ability. Consistent with this result, no differences in the expression of virulence genes were observed among ΔcyaA, ΔfruIΔcyaA, and ΔfruTΔcyaA in either the small intestine of mice or M9 medium supplemented with 20 mM fructose (Supplementary Fig. 2i–l). Taken together, these data suggest that FruI regulates the virulence of *V. cholerae* in the small intestine by downregulating *fruT* expression, which in turn affects the intracellular cAMP level.

**fruI expression is induced by a low O₂ signal in the small intestine via ArcA.** The majority of LacI-type regulators bind to the promoters of carbohydrate metabolic/transport genes in the absence of the specific carbohydrate substrate and repress their expression; this repression is released when the corresponding carbohydrate is available and binds to the carbohydrate-binding domain in the LacI-type regulator[23]. As FruI is a LacI-type regulator, it is expected that the FruI-mediated repression of *fruT* expression would be released in the small intestine, where fructose is present. However, the above finding showed that *fruT* expression was significantly upregualted in ΔfruI compared to that of WT in the small intestine of mice (Fig. 3f). To confirm

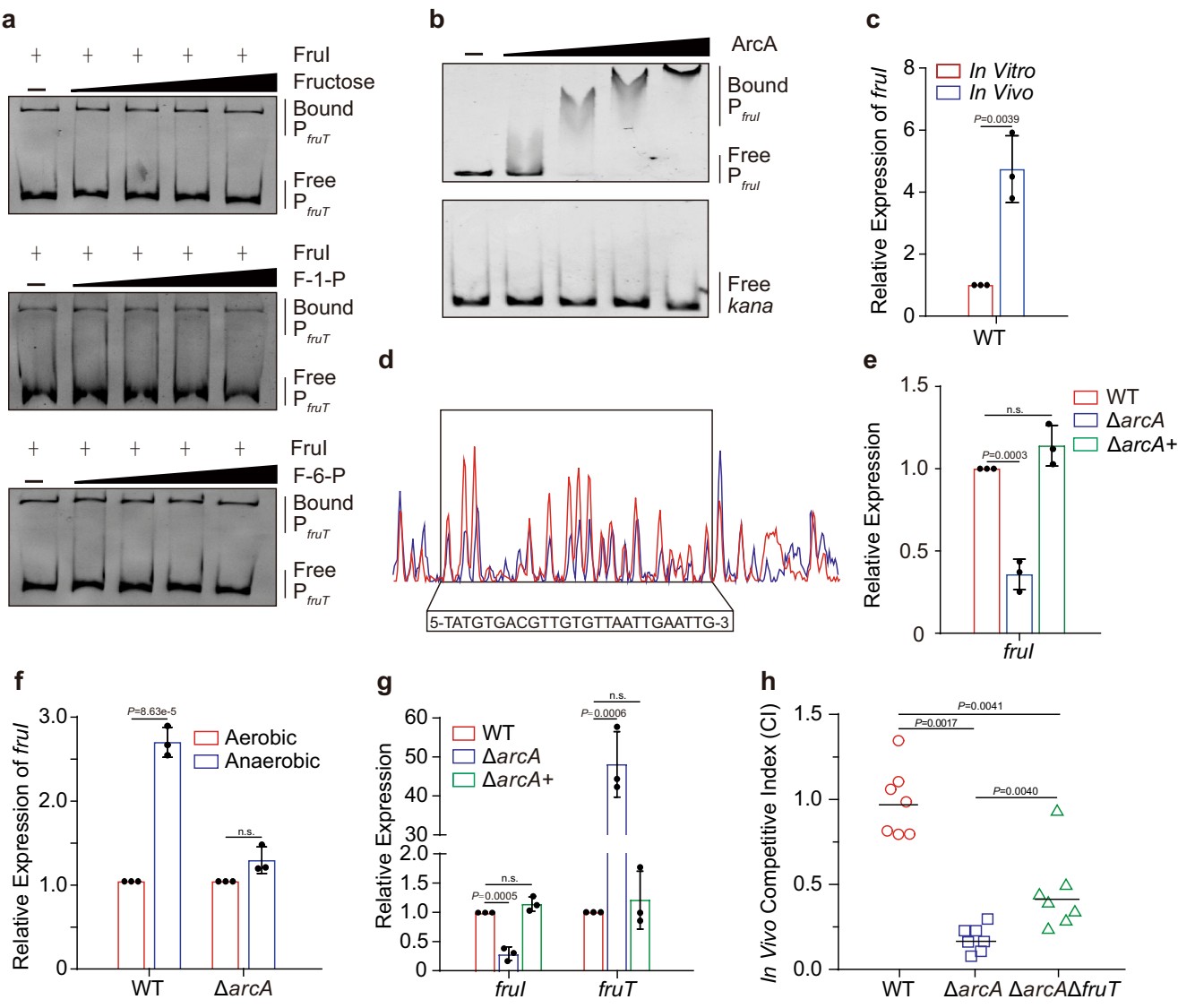

**Fig. 4 *fruI* expression is induced by a low O$_2$ signal in the small intestine via ArcA. a** The influence of fructose, fructose-1-phosphate (F-1-P) and fructose-6-phosphate (F-6-P) on the binding of purified FruI protein to the promoter region of *fruT*. EMSAs of 0.2 μM purified FruI protein complexed with 30 ng of the DNA fragment in the presence of different concentrations of putative effectors (0–10 mM) were conducted. **b** EMSA of the binding of purified ArcA protein to the promoter region of *fruI* and *kana* (negative control). **c** qRT-PCR expression level of *fruI* in WT in the small intestine of mice and M9 medium containing 20 mM fructose as the only carbon source. Data represent the mean ± SD ($n = 3$). **d** ArcA binds to a motif in the *fruI* promoter region. The protected region shows a significantly reduced peak intensities (blue) pattern than seen in compared with those of the control (red). The identified FruI-binding motif is shown in a box at the bottom of the figure. **e** qRT-PCR expression level of *fruI* in WT, Δ*arcA* and Δ*arcA*+ in M9 medium containing 20 mM fructose as the only carbon source. Data represent the mean ± SD ($n = 3$). **f** qRT-PCR expression level of *fruI* in WT and Δ*arcA* in M9 medium containing 20 mM fructose as the only carbon source under anaerobic conditions or aerobic conditions. Data represent the mean ± SD ($n = 3$). **g** qRT-PCR expression level of *fruI* and *fruT* in WT, Δ*arcA*, and Δ*arcA*+ in mouse small intestine. Data represent the mean ± SD ($n = 3$). **h** Competition assay comparing the colonizing ability of WT, Δ*arcA*, and Δ*arcA*Δ*fruT* in infant mouse intestine ($n = 7$ mice per group). CI is defined as the output ratio of mutant strains to WT *lacZ*- divided by the input ratio of mutant strains to WT *lacZ*-. Each symbol represents the CI in an individual mouse; horizontal bars indicate the median. Images are representative of three independent experiments (**a**, **b**). Significance is determined by two-tailed unpaired Student's *t* test (**c**, **e**–**g**) or two-sided Mann–Whitney *U* test (**h**) and indicated as the *P* value; n.s. means no significant difference. Source data are included in Source Data file.

that FruI-mediated repression of *fruT* expression cannot be released by fructose, we also performed EMSAs, and the result showed that the binding of FruI to the *fruT* promoter was not inhibited by fructose and its analogs (Fig. 4a). These data indicate that the FruI-mediated repression of *fruT* expression is not released by fructose and fructose analogs, which would result in FruI-mediated repression of *fruT* expression in the small intestine, where fructose is present.

Since *fruT* regulation by FruI in the small intestine did not appear to depend on the presence or absence of fructose, we

investigated whether it depended on regulation of the expression of *fruI*. To this end, we investigated whether *fruI* expression is induced in *V. cholerae* in the small intestine compared to bacteria incubated in M9 medium supplemented with 20 mM fructose, which mimics the concentration of fructose in the small intestine. qRT-PCR assays showed that *fruI* expression in the WT strain was significantly upregulated in the small intestine of mice compared to that in M9 medium supplemented with 20 mM fructose (Fig. 4c).

To investigate the mechanism that regulates *fruI* expression in the small intestine, we performed DNA pulldown assays to

identify potential regulators that bind to the promoter region of *fruI*. For two regulator proteins (ArcA and Fur) identified, only ArcA specifically binds to the *fruI* promoter, as revealed by EMSAs (Fig. 4b and Supplementary Fig. 4). A ArcA-bound sequence containing an 26–base pair motif (5-TATGT-GACGTTGTGTTAATTGAATTG-3; −67 to −42 from the *fruI* translational start site) in the *fruI* promoter region was revealed by a dye-based DNase I footprinting assay (Fig. 4d), which is similar to the identified ArcA binding consensus sequence[37]. To study whether ArcA actually regulates *fruI* expression, we measured *fruI* expression by qRT-PCR in a Δ*arcA* strain and the WT strain grown in M9 supplemented with 20 mM fructose. We found that *fruI* expression was significantly downregulated in the Δ*arcA* strain and this effect was restored after complementation of Δ*arcA* with *arcA* (Fig. 4e). This indicates that the expression of *fruI* is under direct positive regulation by ArcA.

ArcA and ArcB constitute a two-component regulation system that can sense changes in $O_2$ availability and regulate the expression of downstream genes[38]. Therefore, we studied whether this two-component regulation system regulates *fruI* expression in response to the low $O_2$ concentration in the small intestine[38]. qRT-PCR assay showed that *fruI* expression in the WT strain significantly increased under low $O_2$ conditions compared to high $O_2$ conditions in vitro (Fig. 4f). This effect was not observed in the Δ*arcA* strain (Fig. 4f), indicating that low $O_2$ activates *fruI* expression via ArcA. Furthermore, qRT-PCR assays showed that the expression of *fruI* and *fruT* was significantly decreased and increased, respectively, in Δ*arcA* compared to WT in the small intestine of mice (Fig. 4g). Complementation of Δ*arcA* with *arcA* restored the expression level of *fruI* and *fruT* to the WT level in the small intestine of mice (Fig. 4g). Competitive infection assays in mice showed that compared to the WT, Δ*arcA* exhibited a significant decreased ability to colonize the small intestine of mice (Fig. 4h). These results indicate that *V. cholerae* induces *fruI* expression in the small intestine in response to the low-$O_2$ conditions via ArcB/ArcA, which promotes intestinal colonization of *V. cholerae*.

ArcA is a global transcription regulator in the *Enterobacteriaceae*, which controls the expression of genes involved in several different pathways[39,40]. For instance, ArcA directly regulates the expression of *vpsT* involved in biofilm formation of *V. cholerae*[41], which contributes to bacterial intestinal colonization[42,43]. It is highly likely that the influence of ArcA on the virulence of *V. cholerae* might only be partially dependent of its regulation on *fruI* and *fruT* expression. Competitive infection assays showed that the CI value of the double mutant Δ*arcA*Δ*fruT* versus WT was 0.41 in the small intestine of mice (Fig. 4i), which was significantly higher than the CI value (0.16) of Δ*arcA* versus WT (Fig. 4i), and significantly lower than the CI value (3.18) of Δ*fruT* versus WT (Fig. 3b). These data confirm that in addition to *fruI* and *fruT*, ArcA also influences the virulence of *V. cholerae* via other regulatory pathways.

***V. cholerae* utilizes FruT to take up fructose at low concentrations in seawater**. The above findings suggested that FruT was not used for fructose transport during *V. cholerae* colonization of the small intestine. $H^+$ symporters always operate at low sugar concentrations suggesting that fructose transport by FruT might be important in the seawater where nutrients[44], including carbon sources, are limited. To assess this, we first compared the growth of Δ*fpr* and Δ*fruT* in M9 medium supplemented with 30 mM, 10 mM, 3 μM, or 0.1 μM fructose under aerobic conditions. The results showed that Δ*fruT* cannot grow in medium supplemented with 0.1 μM fructose, which is different from Δ*fpr* and WT (Fig. 5a). It exhibited a growth disadvantage compared to

Δ*fpr* in medium supplemented with 3 μM fructose, and reached a lower final cell counts than Δ*fpr* and WT (Fig. 5b). In contrast, Δ*fpr* exhibited a growth disadvantage in exponential phase compared to Δ*fruT* in medium supplemented with 30 or 10 mM fructose, and it reached almost the same final cell density as that of Δ*fruT* in these two conditions when fructose is almost exhausted (Fig. 5c, d and Supplementary Fig. 5a, b). We also constructed a *fruT*-overexpressing strain (*fruT*++) by introducing a plasmid containing *fruT* into WT, and found that it exhibited a growth advantage compared to WT in medium supplemented with 0.1 or 3 μM fructose, and reached a higher final cell counts than WT (Fig. 5a, b). In contrast, *fruT*++ exhibited almost same growth ability as WT in M9 medium supplemented with 30 or 10 mM fructose (Fig. 5c, d). Furthermore, using fructose uptake assays we found that WT, *fruT*++, and Δ*fpr* exhibited higher fructose uptake efficiency than Δ*fruT* in M9 medium with 0.1 μM fructose (Fig. 5e). In addition, we showed that there was no significant difference in the growth among WT, Δ*fruT*, and *fruT*++ in M9 medium supplemented with 30 mM, 10 mM, 3 μM or 0.1 μM glucose, trehalose, maltose, or ribose (Supplementary Fig. 3a–p). These data indicate that FruT contributes to the uptake of fructose, but not other sugars, at low concentrations.

We then investigated whether FruT contributes to the fructose uptake of *V. cholerae* in seawater. To this end we used samples of seawater from Bohai Sea, which presented fructose concentrations of $0.085 \pm 0.017$ μM (Supplementary Fig. 5c), which is within the ranges previously reported[26]. As natural surface seawater is an aerobic environment[45], fructose uptake assays were performed under aerobic conditions. A total of 54.3%, 54.1%, and 70.5% of the fructose in 500 ml of seawater was utilized by WT, Δ*fpr*, and *fruT*++ in 12 h, respectively, while only 21.8% of the fructose was utilized by Δ*fruT* in 12 h (Supplementary Fig. 5d). These data indicate that *V. cholerae* utilizes mainly FruT to take up fructose at low concentrations in seawater.

**FruT provides a survival advantage to pandemic strains growing in seawater under aerobic conditions**. Since expression of *fruI*, which represses *fruT* expression, is induced by low $O_2$, we hypothesized that in an aerobic surface seawater environment, the repression of *fruT* by FruI is released and that fructose uptake via FruT may benefit the survival of *V. cholerae* in seawater. Indeed, qRT-PCR assays showed that the expression of *fruT* and *fruI* in the WT strain growing aerobically was significantly upregulated and downregulated, respectively, compared to that in the small intestine of mice (Fig. 6a). To test whether *fruT* induction in seawater conferred a survival advantage to *V. cholerae*, we compared the survival ability of WT and Δ*fruT* in seawater under aerobic conditions using competition assays. To mimic the conditions in wide extension of seawater, where the fructose concentration remains constant despite fructose utilization by low abundant microorganisms, WT and Δ*fruT* were located within a membrane-enclosed environment in 500 ml of seawater that was replaced by fresh seawater every 6 h. The CI values of Δ*fruT* versus WT (from 0.211 to 0.009) continued decreasing during a span of 10 days (Fig. 6b). The results indicated that relative to WT, Δ*fruT* exhibited a substantial survival disadvantage when growing aerobically in seawater. The similar result was obtained when competition assays was performed in artificial seawater medium that was supplemented with 0.1 μM fructose as the only carbon source every 6 h under aerobic conditions (Fig. 6c). However, WT almost competed evenly with Δ*fruT* in artificial seawater medium supplemented with 0.1 μM glucose as the only carbon source every 6 h under aerobic conditions (Fig. 6d), confirming that FruT confers pandemic *V. cholerae* a survival advantage in seawater through enabling bacteria to efficiently uptake

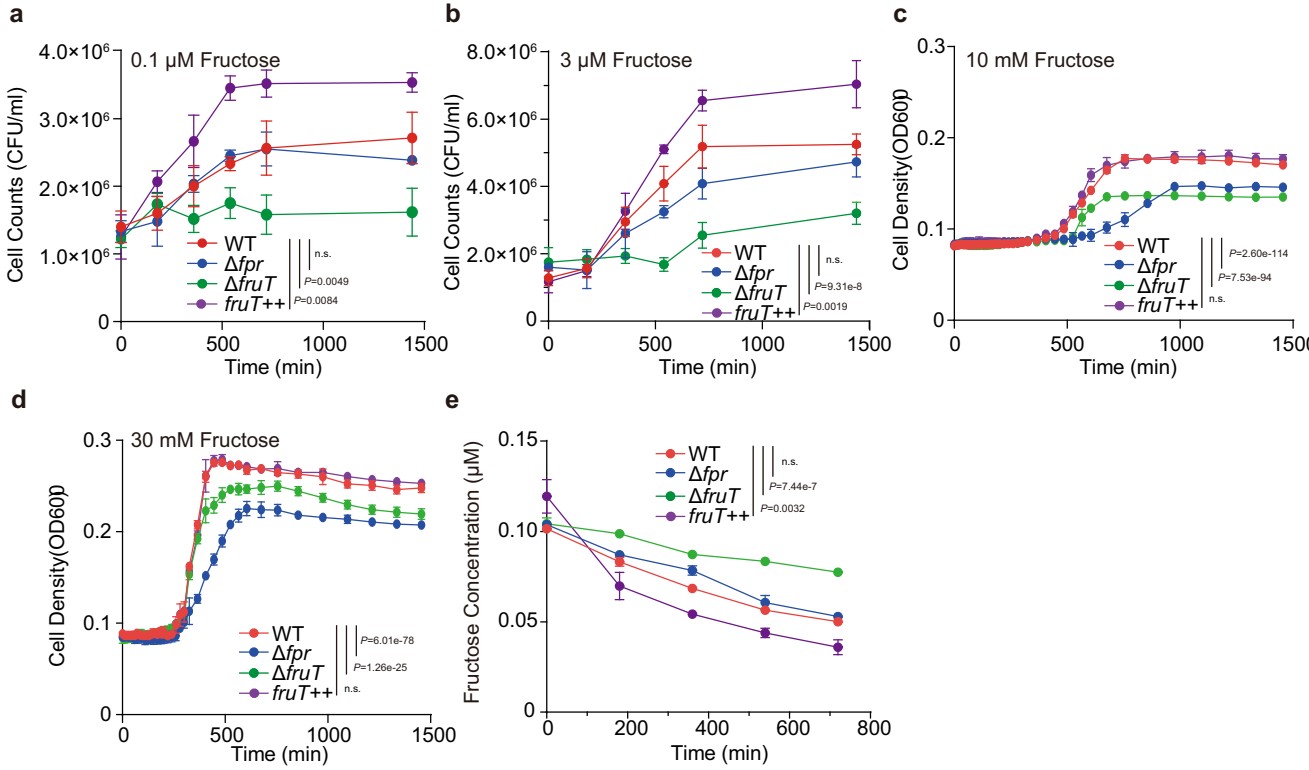

**Fig. 5 FruT is responsible for the bacterial uptake of fructose at low concentrations in seawater. a–d** Growth curves of WT, $\Delta fruT$, $\Delta fpr$, and $fruT++$ in M9 medium containing 0.1 μM (**a**), 3 μM (**b**), 10 mM (**c**), 30 mM (**d**) fructose as the only carbon source under aerobic conditions. Cell growth was followed by plating and determining the cell counts on LB agar plates (**a**, **b**) or measuring the absorbance at 600 nm (**c**, **d**). Data represent the mean ± SD ($n = 3$). **e** Fructose consumption by WT, $\Delta fruT$, $\Delta fpr$, and $fruT++$ grown in M9 medium containing 0.1 μM fructose as the only carbon source. Data represent the mean ± SD ($n = 3$). Significance is determined by two-way ANOVA and indicated as the $P$ value; n.s. means no significant difference. Source data are included in Source Data file.

low-concentration fructose under aerobic conditions. In contrast, $\Delta fruT$ exhibited the same survival ability as WT when growing anaerobically in artificial seawater medium supplemented with 0.1 μM fructose as the only carbon source every 6 h (Fig. 6e).

Overall, these results indicate that in aerobic surface seawater environment, the repression of $fruT$ by FruI is released and that FruT-mediated fructose uptake plays an important role in the survival of *V. cholerae* in aerobic seawater environments.

## Discussion

Pandemic *V. cholerae* lives in aquatic environment and causes diseases in human small intestine. Here, we elucidated the molecular mechanisms that enable pandemic *V. cholerae* strains to use a fructose/H$^+$ symporter FruT to uptake low-concentration fructose in seawater, conferring them a survival advantage in that nutrient-limited environment. Pandemic *V. cholerae* turns off $fruT$ via FruI and uptake high-concentration fructose via PTS in the mouse small intestine, resulting in the decrease of intracellular cAMP level and upregulation of virulence gene expression, and maintaining high pathogenicity in vivo.

*V. cholerae* spends much of its life cycle outside of the human host in aquatic environment, such as seawater, and must acquire nutrients from its environment[6]. The bioavailability of carbon sources is limited in seawater[7]. In this study, we showed that acquiring the fructose/H$^+$ symporter FruT enables pandemic *V. cholerae* strains to efficiently take up fructose at low concentrations in seawater. There are other sugars present in seawater; however, their concentrations (0.01–0.24 μM) are much lower than that of fructose[26], except for glucose, whose concentration (0.03–0.67 μM)

is similar to that of fructose[26]. Future research is needed to determine, in addition to glucose PTS, whether pandemic *V. cholerae* strains have a transporter specific for uptaking low-concentration glucose.

LacI-type regulators are widely used in bacteria for control of carbon source utilization in response to different environments. Through comparative genomics analysis, we showed that, among 11 genes encoding LacI-type regulators in *V. cholerae*, $fruI$ is significantly associated with pandemic strains. $fruT$, whose expression is under negative regulation by $fruI$, is located 1.3 kb upstream of $fruI$. Comparative genomics analysis showed that the distribution of $fruT$ in pandemic and nonpandemic *V. cholerae* is identical to that of $fruI$, indicating $fruI$ and $fruT$ were introduced into *V. cholerae* genome via a single lateral gene transfer event. As pandemic *V. cholerae* needs to repress $fruT$ expression using FruI in the small intestine of mice, which benefits the virulence gene expression and bacterial colonizing ability in vivo, simultaneous acquisition of $fruI$ and $fruT$ enabled pandemic *V. cholerae* strains to obtain a survival advantage in seawater while maintaining high pathogenicity in animal intestine.

Both $fruT$ and $fruI$ are located within a predicted genomic island (GI-7). GIs are clusters of genes that are mobile via mechanisms of lateral gene transfer[46]. Thus, it is highly likely that $fruI$ and $fruT$ were introduced into *V. cholerae* genome via a single lateral gene transfer event. Over 400 potential GIs have been predicted in different pandemic or nonpandemic *V. cholerae* strains[47]. However, the role of most of these GIs in the evolution of *V. cholerae* remains unclear, with slight exceptions that are related to virulence or antibiotic resistance, such as VPI that encodes toxin-coregulated pilus, VSP-I that includes a gene

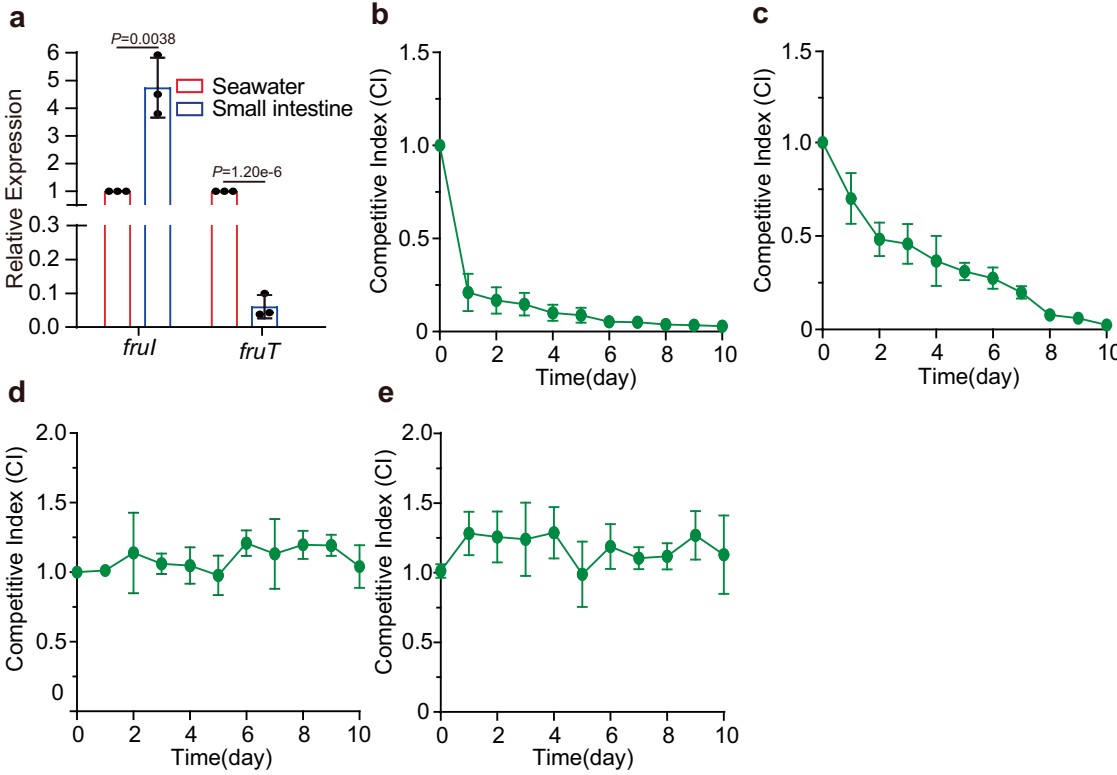

**Fig. 6 *V. cholerae* turns on FruT in response to O$_2$ for growth in seawater. a** qRT-PCR expression level of *fruI* and *fruT* in the mouse small intestine and seawater under aerobic conditions. Data represent the mean ± SD ($n = 3$). **b** Competition assay comparing the survival ability of WT and Δ*fruT* within a membrane-enclosed environment in seawater that was replaced by fresh seawater every 6 h under aerobic conditions. **c**, **d** Competition assay comparing the survival ability of WT and Δ*fruT* located within a membrane-enclosed environment in artificial seawater medium that was supplemented with 0.1 μM fructose (**c**) or 0.1 μM glucose (**d**) as the only carbon source every 6 h under aerobic conditions. **e** Competition assay comparing the survival ability of WT and Δ*fruT* located within a membrane-enclosed environment in artificial seawater medium that was supplemented with 0.1 μM fructose under anaerobic conditions. CI was determined as the output ratio of Δ*fruT* to WT *lacZ*- cells divided by the input ratio of Δ*fruT* to WT *lacZ*- cells. Data represent the mean ± SD ($n = 3$). Significance is determined by two-tailed unpaired Student's *t* test (**a**) and indicated as the *P* value; n.s. means no significant difference. Source data are included in Source Data file.

encoding a dinucleotide cyclase required for bacterial intestinal colonization, and SXT that carries antibiotic resistance genes[48,49]. In this study, we provided evidence that *fruI* and *fruT* in GI-7 give *V. cholerae* advantages for survival in seawater and maintain bacterial pathogenicity in animal intestine. In addition to *fruI* and *fruT*, there are other four genes in GI-7, whose function are unknown, and whether they are related to the virulence or environmental survival of *V. cholerae* needs to be investigated in future studies.

cAMP is a universal second messenger that is used by diverse forms of life, and it has central roles in regulating gene expression related with many biological process, such as carbon metabolism, virulence, and biofilm formation, in bacteria[36]. In *V. cholerae*, cAMP-CRP represses the expression of several virulence genes, such as *ctxAB* encoding the cholera toxin and *tcpA* encoding the repeating subunit of toxin-coregulated pilus, via influencing the expression of genes encoding regulator TcpP and ToxR[19–21]. cAMP-CRP also represses the biofilm formation of *V. cholerae* by regulating the expression of a set of genes, such as *vpsR* encoding a regulator and *cdgA* encoding a cyclic diguanylate cyclase that promotes the production of cyclic di-GMP[50]. cAMP-Crp was also reported to contribute to the resistance of *V. cholerae* to environmental bacteriophages[51]. PTS is a major factor involved in the regulation of intracellular cAMP level in bacteria[52], and uptake of sugar via PTS results in the decrease of bacterial intracellular cAMP level[18,36]. PTS components play multiple regulatory roles in enteric bacteria[53]. In *V. cholerae*, EIIA of glucose PTS

influences biofilm formation via modulating the activity of cyclic di-GMP phosphodiesterase PdeS and regulating the decay of CsrB/C sRNA[54,55]. In this study, we showed that in response to the low O$_2$ signal in animal small intestine, *V. cholerae* turns off *fruT* and uptakes fructose via PTS, leading to the decrease of intracellular cAMP level and the enhanced expression of virulence genes. It is likely that in addition to inducing virulence gene expression found in this study, the elevated fructose PTS activity of *V. cholerae* in animal small intestine also influences bacterial pathogenicity via other mechanisms, such as regulating the formation of biofilm.

Compensatory relationships have been observed between glucose transporters in mammalian cells; however, the underlying mechanisms are unclear[56]. There is a previously known fructose PTS in *V. cholerae*[17]. In this work, we identified a fructose/H$^+$ symporter FruT in *V. cholerae*, and showed that the repression of *fruT* expression by FruI enhances the expression of fructose PTS. Furthermore, qRT-PCR assays showed that the deletion of *fpr* encoding a component of fructose PTS led to the elevated expression of *fruT*, but not *fruI*, in M9 medium supplemented with 20 mM fructose, while the expression of both *fruT* and *fruI* in Δ*fpr* exhibited no significant change compared to that of WT in M9 medium supplemented with 20 mM glucose (Supplementary Fig. 6a, b), indicating fructose PTS also influences the expression of *fruT* via modulating fructose uptake. It is highly likely that there is a compensatory relationship between FruT and fructose PTS. The repression of either of these two fructose

transports leads to the increased expression of the other transporter. The exact compensatory mechanism between FruT and fructose PTS is currently unclear; however, our data showed that FruI does not play a role in that mechanism. We hypothesize that the reciprocal regulation between fructose PTS and FruT maybe mediated via the change of bacterial intracellular fructose concentration, which will be the subject of future studies.

## Methods

**Bacterial strains, plasmids, and growth conditions**. *V. cholerae* O1 El Tor strain El2382 isolated in 1994 was provided by the Shanghai Municipal Centers for Disease Control & Prevention. *Escherichia coli* DH5a/λpir strain was used for DNA manipulation, and *E. coli* S17/λpir strain was used for conjugation with *V. cholerae*. Bacterial strains were grown in Luria-Bertani (LB) broth, M9 medium, or artificial seawater medium (Sigma; S9883) with different concentrations of glucose, fructose, trehalose, maltose, or ribose and desterilized seawater from the Bohai Sea, respectively. Bacteria were inoculated into fresh medium and grown at 37 °C with shaking (for aerobic conditions) or without shaking (for anaerobic conditions). Antibiotics (Sigma) were used at the following concentrations, unless otherwise indicated: polymyxin B, 40 μg/ml; ampicillin, 50 μg/ml; chloramphenicol, 30 μg/ml (for *E. coli*) or 5 μg/ml (for *V. cholerae*).

**Mutant construction, complementation, and overexpression**. All primers used in this study can be found in Supplementary Table 2. For mutant construction, the DNA fragments were cloned into the suicide vector pRE112[57]. Plasmids were maintained in *E. coli* S17-1/λpir for conjugation to *V. cholerae*. Integration was selected on LB plates containing polymyxin B (40 μg/ml) and 20% sucrose for *sacB* counterselection. PCR and DNA sequencing were used to confirm the presence of the induced mutations. For complementation, genes with native promoter were amplified by PCR and cloned into vector pBAD33[58]. The complementary plasmids were introduced into *V. cholerae* via electroporation. For overexpression of *fruT*, *fruT* was amplified by PCR and cloned into vector pTrc99a[59]. The plasmid was introduced into *V. cholerae* via electroporation.

**RNA isolation and RNA sequencing**. *V. cholerae* were harvested from small intestinal tissue or after culturing at 37 °C for LB broth or M9 medium or at 22 °C for seawater samples. Total RNA was isolated using TRIzol Reagent (Invitrogen; 15596026) according to the manufacturer's protocol. RNA was quantified and qualified using an Agilent 2100 Bioanalyzer (Agilent Technologies), a NanoDrop (Thermo Fisher Scientific Inc), and 1% agarose gel electrophoresis. One microgram of total RNA with an RIN value >6.5 was used for library preparation. rRNA (including 16S and 23S rRNA) was depleted from total RNA using Ribo-off rRNA Depletion Kit (Bacteria) (Vazyme; N407-01). Libraries were constructed and analyzed by NOVOGENE, Inc (TianJin, China). Differentially expressed genes in Δ*fruI* compared to WT were identified using edgeR (|fold change| > 2 and P < 0.05). P values were adjusted using the Benjamini and Hochberg approach to control the false discovery rate.

**qRT-PCR**. qRT-PCR analysis was conducted on Applied Biosystems ABI 7500 (Applied Biosystems) with SYBR green fluorescence dye. cDNA templates were denatured at 95 °C for 10 min, followed by 45 cycles of 95 °C (30 s) and 60 °C (60 s). *rrsA* gene was used as a reference control for sample normalization[60], and the relative expression level was calculated as fold change values using the $2^{-\Delta\Delta CT}$ method[61]. Each experiment was carried out in triplicate.

**Growth curve**. To determine the growth curve of each strain in 10 or 30 mM sugars, overnight cultures were washed with PBS three times and diluted to $10^6$/ml in a flask containing 20 ml of M9 medium with 10 or 30 mM fructose, glucose, trehalose, maltose, or ribose. When necessary, IPTG was added to a final concentration of 0.1 mM. A 200 μl aliquot was added to a 96-well microplate and incubated at 37 °C with shaking at 180 rpm for 24 h. The absorbance at 600 nm was recorded. Experiments were independently performed three times.

To determine the growth curve of each strain in 0.1 or 3 μM sugars, overnight cultures were washed by PBS for three times and diluted to $10^6$/ml in a flask containing 200 ml of M9 medium containing 0.1 or 3 μM fructose, glucose, trehalose, maltose, or ribose and incubated at 37 °C with shaking at 180 rpm. A 100 μl aliquot was removed from the flask and suitable dilutions were plated on LB agar plates. The growth rate was determined by cell counts every 3 h. Experiments were independently performed three times.

**EMSA**. The 6×His-tagged FruI, ArcA, and Fur proteins were expressed and purified in *E. coli* BL21 (DE3). DNA target fragments were amplified by PCR and purified using a SPARKeasy Gel DNA Extraction Kit (Sparkjade; AE0101). Purified PCR fragments (40 ng) were incubated at 25 °C for 30 min with 6×His-tagged FruI or ArcA protein at concentrations ranging from 0 to 2 μM in 20 μl reactions containing Binding buffer (10 mM Tris-HCl [pH 7.5], 0.2 mM dithiothreitol, 5 mM MgCl₂, 10 mM KCl, and 10% glycerol). When necessary, 30 mM acetyl phosphate

was added to the binding buffer before incubation. The protein-DNA fragments were electrophoretically separated on a native polyacrylamide gel at 4 °C and 80 V/cm. The gel was stained for 10 min in a solution of 0.1% GelRed (Biotium; 41003), and protein bands were visualized by ultraviolet transillumination. During effector screening studies, the purified 6×His-tagged FruI protein was incubated at 25 °C for 10 min with 0–10 mM fructose, fructose-1-phosphate, or fructose-6-phosphate, before being added to binding buffer.

**Dye primer-based DNase I footprinting assay**. About 200-bp fragment of the *fruI* or *fruT* promoter regions was generated by PCR with 6-FAM primers. Forty nanograms of 6-FAM-labeled *fruI* or *fruT* promoter was incubated with different amounts of ArcA or FruI protein ranging from 0 to 1 μM in a binding buffer (10 mM Tris-HCl [pH 7.5], 0.2 mM dithiothreitol, 5 mM MgCl₂, 10 mM KCl, and 10% glycerol). 0.05U DNase I (Thermo; EN0521) was added to a 20-μl reaction for 5 min at room temperature. The reaction was stopped by heating at 65 °C for 10 min in the presence of 250 mM ethylenediaminetetraacetic acid (EDTA). DNA fragments were purified with the QIAquick PCR Purification kit (Qiagen; 28104) and eluted in 15 μl distilled water. Samples were analyzed by MAP Biotech CO., Ltd (Shanghai China). The results were analyzed with Peak Scanner (v1.0).

**Intestinal colonization assay**. Both sexes of CD-1 infant mice (5 days old) were purchased from the Beijing Vital River Laboratory Animal Technology (Beijing, China). Mice were housed under specific pathogen-free conditions with a 12 h light/dark cycle, at a temperature of 30 °C incubator, and a relative humidity of 50 ± 5%. All animal studies were conducted according to protocols approved by the Institutional Animal Care Committee of Nankai University (Tianjin, China) and performed under protocol no. IACUC 2016030502. Briefly, the *V. cholerae* lacZ+ strains (WT and the mutants) and lacZ− strains (ΔlacZ) were grown overnight at 37 °C in LB broth. Approximately $10^5$ lacZ+ strains were mixed with an equal amount of WT lacZ− strain and the mixtures were intragastrically administered to groups of six anesthetized mice. After 24 h of incubation, the small intestine was removed, homogenized, and plated onto LB agar plates containing 5-bromo-4-chloro-3-indoyl-β-d-galactopyranoside (X-gal) or chloramphenicol (5 μg/ml) to enumerate the recovered bacteria and to obtain the output ratios. The CI was determined as the output ratio of lacZ+ to lacZ− cells divided by the input ratio of lacZ+ to lacZ− cells. The CI of complemented strains was determined as the output ratio of lacZ+(pBAD33) to lacZ−(pBAD33) cells divided by the input ratio of lacZ+(pBAD33) to lacZ−(pBAD33) cells.

**H⁺ symport assay**. The detection of H⁺ movements associated with initial sugar uptake was assessed by adding sugar pulses to unbuffered cell suspensions[62]. Bacteria (OD₆₀₀ of 1.0) were harvested by centrifugation (5000 g, 5 min, 4 °C), washed with ice-cold PBS, resuspended in 0.9% NaCl to a final concentration of $10^6$/ml and kept in 37 °C for 3 h for starvation. After starvation, 500 μl of WT, mutants, complement strains, and 0.9% NaCl (negative control) were seeded into a 24-well plate (Seahorse XF24 Cell Culture Microplates, Agilent). After acquiring baseline measurements, the following drug was used for an extracellular pH test: 30 mM fructose (port A). Cells were assessed for extracellular alkalinization rate using the Seahorse XF analyzer (Agilent)[63].

**Determination of the Fpr phosphorylation state**. A total of 0.2 ml of bacterial culture (OD₆₀₀ of 1.0) was treated with the addition of 20 μl of 10 M NaOH followed by 1 ml of ethanol and 180 μl of 3 M sodium acetate, pH 5.2. After chilling at –80 °C for 2 h, precipitates were collected by centrifugation, rinsed with 70% ethanol, and suspended in 100 μl of sample buffer (160 mM Tris-HCl, pH 7.5, 4% SDS, 20% glycerol, and 10% 2-mercaptoethanol). To achieve a good separation of the two forms of Fpr, samples were fractionated on SDS-PAGE gels containing 50 μM of Phos-tag (WAKO; 304-93521) and 40 μM MnCl₂. Subsequently, gels were washed with western transfer buffer (25 mM Tris, 192 mM glycine, 20% methanol, and 0.1% SDS) containing 5 mM EDTA for 10 min, followed by a second wash with transfer buffer for 20 min.

**Western blotting**. Bacterial cells were harvested and washed three times in ice-cold PBS, re-suspend by ice-cold PBS, and sonicated with 15 cycles of 30 s on/off at 95% power. Then the cell debris was removed by centrifuging at 12,000 g for 10 min at 4 °C. The supernatants were collected and kept on ice before use. Total cellular proteins (20 ug) were separated by SDS-PAGE and transferred to 0.2 μm polyvinylidene difluoride membranes (Bio-Rad) by electroblotting. Blots for Fpr, RNAP (RNA polymerase), and cholera toxin protein used the anti-Fpr monoclonal antibody (Willget Biotech Co., Ltd.), anti-RNA polymerase beta antibody (ab191598), and anti-cholera toxin antibody (ab123129) at 1:2000 dilution, respectively. Western blotting was detected using horseradish peroxidase-linked goat anti-rabbit IgG secondary antibodies (1:5000 dilution, Sparkjade; EF0002) and Sparkjade ECL plus (Sparkjade; ED0016) as recommended by the manufacturer. Images were acquired using an Amersham™ Imager 600 System (General Electric Company). Western blotting bands were quantified using Image J (v1.8.0) software.

**DNA pulldown assay**. The *fruI* promoter regions were amplified by 5'biotin tagged primers. The negative control DNA fragments were amplified in the *fruI* coding region. The biotin-labeled DNA fragments were purified using a SPAR-Keasy Gel DNA Extraction Kit and diluted to a final concentration of 30 ng/µl.

Two hundred microliter of Streptavidin-agarose beads (Thermo Fisher) was washed three times in 2×B/W buffer (10 mM Tris-HCl pH 7.5, 1 mM EDTA, 2 M NaCl) and incubated for 1 h with an equal volume of biotin-labeled DNA fragments at room temperature. DNA-treated beads were washed three times in BS-THES buffer (5×BS buffer: 50 mM HEPES, 25 mM CaCl2, 250 mM KCl, 60% glycerol, 1.5% protease inhibitor, 0.2% phosphatase inhibitor; 2.25×THES: 50 mM Tris-HCl pH 7.5, 10 mM EDTA, 20% sucrose, 140 mM NaCl). The overnight cultures were diluted 1:100 in a flask containing 100 ml fresh LB broth and grown to log phase (OD$_{600}$ of 0.6–0.8). Bacteria were harvested and washed three times in ice-cold PBS and re-suspend by BS-THES buffer. Cells were sonicated with 15 cycles of 30 s on/off at 95% power. Then the cell debris was removed by centrifuging at 12,000 g for 10 min at 4 °C. The supernatants were collected and kept on ice before use. DNA-treated Beads were incubated in 1 ml supernatants twice for 30 min at room temperature. The beads were then washed three times with BS/THES. Protein-DNA complexes were eluted by Elution buffer (25 mM Tris-HCl pH 7.5, 300 mM NaCl) and further treated with 20% (v/v) trichloroacetic acid in ice for 3 h to precipitate protein. The precipitates were collected by centrifugation at 12,000 g for 10 min at 4 °C. The pellets protein were segregated by SDS-PAGE and the specific bands were collected and analyzed by liquid chromatography-tandem mass spectrometry in Beijing Genomics Institution (Beijing, China).

**Intracellular cAMP assay**. To determine the intracellular cAMP level of each strain, WT and mutants were harvested by centrifugation (5000 g, 5 min, 4 °C), weighed, washed with ice-cold PBS, resuspend in 100 mM HCl, and ultra-sonicated with 15 cycles of 30 s on/off at 95%. Cell debris was removed, and the resulting supernatant was used to determine the cAMP level. Intracellular cAMP level was determined using a competitive cAMP ELISA kit (abcam133051) following the manufacturer's instructions.

**Fructose uptake assay**. Overnight cultures were diluted to 10$^6$/ml of M9 broth with 0.1 µM fructose or seawater without antibiotics and incubated at 37 or 22 °C with shaking at 180 rpm. When necessary, IPTG was added to a final concentration of 0.1 mM. A 100 µl aliquot was removed from the flask every 3 h, and the concentration of fructose was measured by a PicoProbe Fructose Fluorometric Assay Kit (BioVision; K611) following the manufacturer's instructions.

**Fructose concentration detection in seawater**. In total, eight randomly selected water samples from the Bohai Sea were tested. Seawater samples were sterilized by immediately passing through a 0.22-micron bacteria-retentive filter. Then, the samples were frozen before detection. The concentration of fructose was measured by a PicoProbe Fructose Fluorometric Assay Kit.

**Survival assay**. Collected seawater was prefiltered through a 50-µm quartz filtration system and then filtered through 0.22-µm pore-size membranes (Millipore). The overnight cultures were washed and diluted to 10$^6$/ml by sterilized seawater, and approximately 10$^6$ Δ*fruT* was mixed with an equal amount of WT *lacZ*− strain. The mixtures were located within a membrane-enclosed environment in 500 ml seawater or in artificial seawater medium. The seawater was replaced with fresh seawater and the artificial seawater medium was supplemented with 0.1 µM fructose or 0.1 µM glucose as the only carbon source every 6 h. The sample were incubated at 22 °C with shaking (for aerobic conditions) or without shaking (for anaerobic conditions). A 100 µl aliquot was removed from the membrane-enclosed environment every 24 h, and suitable dilutions were plated onto LB agar plates containing X-gal to enumerate the recovered bacteria and to obtain the output ratios. The CI was determined as the output ratio of Δ*fruT* to WT *lacZ*− cells.

**Statistical analysis**. All data are expressed as the means ± standard deviation (SD). Differences between two groups were evaluated using two-tailed unpaired Student's *t* test, Mann–Whitney *U* test, or two-away ANOVA according to the test requirements (as stated in the figure legends). Significance is indicated as the *P* value and n.s. means no significant difference. Statistical analysis of Mann–Whitney *U* test was conducted using the software MedCalc (v12.3.0.0). Statistical analysis of two-tailed unpaired Student's *t* test was conducted using the Microsoft Excel 2019 (16.0.14026.20294). Statistical analysis of two-way ANOVA was conducted using the software using GraphPad.Prism.7.0.4 (GraphPad Inc., San Diego, CA).

**Reporting summary**. Further information on research design is available in the Nature Research Reporting Summary linked to this article.

## Data availability

The RNA-seq data have been deposited in the NCBI Sequence Read Archive database under accession code SRR12474047 and SRR12474048. DNase I footprinting assay data have been deposited in OSF under accession code 4SQ62. Other source data are provided with this paper in the Source Data file. Source data are provided with this paper.

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

## Acknowledgements

This work was funded by National Key Programs for Infectious Diseases of China Grant 2017ZX10303405-001 (to L. W.), National Natural Science Foundation of China Grant 31820103002 (to L. W.), 31530083 (to L. W.), 31770144 (to L. W.), 81871624 (to L. F.), 81772148 (to B. L.), 81902030 (to T. X.), and National Key R&D Program of China Grant 2018YFA0901000 (to L. F.).

## Author contributions

L.W. and L.F. designed the research; Y.L., T.X., Q.W., W.L., J.W., X.Z., B.L., R.L., and X.L. performed the research; L.F., B.L., W.L., and T.X. contributed new reagents or analytic tools; Y.L., B.L., T.X., Q.W., X.G., and L.W. analyzed the data; L.W., B.L., Y.L., and L.F. wrote the manuscript.

## Competing interests

The authors declare no competing interests.
