## [Peer Review File · Nature Communications]

REVIEWER COMMENTS

Reviewer #1 (Remarks to the Author):

The paper exams the use of a PTS sugar by *V. cholerae* and specifically claim fructose uptake is essential for *V. cholerae* survival in seawater. Overall, the experiments are well done and appropriate controls were used, the results section is sound in their execution. However, the papers introduction and discussion are weak and the overall writing of the paper needs work. The writing needs a lot of work.

Major comments

Spelling out *V. cholerae* throughout the paper, Vc not appropriate.

Line 140-142. I do not know how this data shows VCA0673 regulates fructose

Line 147 fructose levels 30mM are very high in these experiments should be 10 mM.

Line 331 fructose concentration stated as 20mM

Other sugars should be tested

Referencing in general is very poor and many references are not appropriate or incorrect. Many statement of fact are not referenced.

Line 54. Worden AZ not an appropriate reference for this statement

Line 55. Faruque SM discuss virulence genes among environmental strain, this paper is a review and does not examine persistence

Line 58 Reference not suitable, this paper measure DOC from a single water sample from the Netherlands, not representative of seawater.

Line 61 this statement is not true, many studies have examined nutrient uptake by *V. cholerae* and similarly lines 433-437.

Line 66-69 needs a reference

Line 209-110. I could not find this reference and since the paper hinges on the assumption that fructose is limiting in seawater as opposed to other sugars. Also is this the only paper to show this from 1980?

Line 440-445. Not sure how this relates to the present study and the data described

Line 460 what is GI-7, this is the first mention. Also the whole paragraph is out of place. No mention of VPI-1 or VPI-2 and what they encode and how they related to virulence and metabolism

Line 472 No mention of previously described work on CRP and cAMP or PTS sugars in general in *Vc*. Significant work has been performed by others?

Line reference 24 should ready available.

Reviewer #2 (Remarks to the Author):

In the manuscript by Liu et al, the authors characterize a novel fructose utilization locus in pandemic clones of *V. cholerae* composed of a repressor, FruI, and transporter, FruT. The authors demonstrate that FruT aids growth of *V. cholerae* in the marine environment, which is perhaps the most convincing data in the manuscript. They also show that the expression of FruT during infection may marginally reduce the fitness of *V. cholerae* in vivo. They show that the presence of FruT slightly alters virulence gene expression and attempt to mechanistically tie this observation to modulation of cAMP levels, however, the data presented do not provide a strong link between these phenotypes. Furthermore, the authors identify that the O₂-sensing regulator ArcA activates expression of FruI under anaerobic conditions to prevent FruT expression. The data to link ArcA-dependent regulation of FruT to the phenotypes observed in vivo, however, are not convincing. While the study is clearly written, there are a number of key controls or comparisons missing that make it difficult to validate the model presented. If these are addressed, I believe this manuscript could make a valuable addition to the field. Specific comments, questions, and suggestions for the authors to consider are listed below.

Fig. 1C – these differences are very subtle. Unless they are statistically significant it seems inappropriate to comment on them. Or at the very least, statements about these differences in the text must be qualified to make it clear that they are not statistically significant.

Previous studies have shown that deletion of Fpr and/or EI prevents growth/fermentation of El Tor V cholerae on fructose (ref 14 and 15). This contrasts with the results obtained in Fig. 1D, which show that a Δfpr strain can still grow on fructose. Can the authors comment on this discrepancy? Does an EI mutant exhibit a similar degree of growth on fructose as the Fpr mutant?

Fig. 2 – are these differences statistically significant? Error bars seem to overlap between samples in this experiment, although it is unclear what these error bars signify. This information should be added to the figure legend. Statistical comparisons should be performed and commented on in the text.

Why wasn't the $\Delta fruI \Delta fruT$ mutant competed against the parent strain in Fig. 3C? I believe that the expected outcome based on the authors model would be a mutant that phenocopies the $\Delta fruT$ mutant. The comparison made in Fig. 3C is fine, however, it requires an additional and unnecessary logical step by competing this strain against a $\Delta FruT$ parent. Which leaves me wondering whether competition between the lacZ- parent and the $\Delta fruI \Delta fruT$ strain did not show the expected phenotype.

Fig. 3E – relative differences in transcript abundance are subtle for all of the assays in Fig 3. Also, the in vitro growth conditions used in 3E would not be expected to induce the expression of virulence genes in an El Tor V cholerae strain. El Tor strains require growth in AKI medium to successfully activate the virulence regulatory program. As a result, these data should be further supported via Western blot analysis to show that the expression of key virulence proteins are altered to mirror the changes observed in transcript abundance. Detection of cholera toxin or TcpA is commonly used in the field.

Fig. 3H and 3I. Transcript abundance of Fpr is supposed to be elevated ~2-fold in $\Delta fruT$ according to 3H, however, the protein abundance for Fpr does not increase 2-fold in 3I. If anything, this blot would indicate that protein abundance for Fpr might actually be decreased in $\Delta fruT$. Although this comparison is difficult to make because a loading control is not included. Regardless, the relevance of the transcript abundance changes observed are unclear. Based on the increase in unphosphorylated Fpr observed in the $\Delta fruT$ mutant, it appears as though FruT does have some regulatory effect of fructose transport through the PTS. This is assuming that the relative changes in the abundance of unphosphorylated Fpr to phosphorylated Fpr observed are reproducible and significantly different based on quantification of band intensities, which has not been performed. But FruT influencing PTS transport through regulation of Fpr expression does not seem to be the underlying mechanism.

The authors demonstrate that FruI and FruT may impact PTS transport of fructose through deletion of those factors. But does the PTS influence FruI/FruT expression or activity?

Throughout the study, mutant strains are largely studied in minimal medium with fructose as a sole carbon source. A relevant negative control condition would be beneficial in many instances to demonstrate that the phenotypes observed are specific to growth on fructose. Growth in minimal medium with glucose (PTS transported) or maltose (ABC transported) could be relevant negative controls. For example, in Fig. 3I it would be valuable to examine samples grown in something other than fructose to ensure that the non-phosphorylated Fpr observed is a direct result of fructose uptake through the PTS as hypothesized. Similarly, transcript abundance of virulence genes for strains grown in glucose would help support the conclusion that differences observed are specific to fructose uptake by FruT as hypothesized. A control like this is actually included in Fig. 6D, and is highly valuable in the context of that experiment.

The link between cAMP and virulence gene regulation with regard to fruT and fruI is largely correlative and/or based on epistasis between fruI and cyaA in vivo (Fig. S2), the latter of which could be influenced by pleiotropic effects of the cyaA deletion in vivo. If the effect of fruT on virulence gene expression occurs via altered cAMP levels as the authors hypothesize, then addition of extracellular cAMP at physiological levels should reduce the enhanced virulence gene expression observed in the $\Delta fruT$ mutant. Also, deleting cyaA (or CRP) in the parent, $\Delta fruI$, and $\Delta fruT$ backgrounds (to make double mutants) should ablate differences in virulence gene expression in

in vitro under conditions similar to those tested in Fig. 3E.

Expression of FruT is relatively low in vitro when FruI is intact according to Fig. 1. This may explain why the phenotypes to demonstrate that FruT supports growth on low fructose concentrations in Fig. 5 are very subtle. FruT expression is greatly increased in a $\Delta fruI$ mutant. So, to more convincingly demonstrate that FruT is really a fructose transporter that can promote fructose uptake at low concentrations it might be valuable to assess the fructose uptake and growth of a FruI mutant in experiments that parallel those shown in Fig. 5A-E. Testing a $\Delta fruI$ mutant would also be valuable in Fig. 2 for the same reason. Or alternatively, simply overexpressing FruT from an inducible promoter might be even better because it would avoid the pleiotropic changes in gene expression that occur in the $\Delta fruI$ mutant.

Fig. 5E – are these data representative of multiple experiments? Are the differences statistically significant?

Deletion of ArcA may attenuate *V. cholerae* in vivo due to the derepression of FruT as hypothesized. However, it is equally likely that ArcA may have a distinct effect on *V. cholerae* virulence that is entirely independent of FruT expression. To formally test this, a $\Delta fruT \Delta arcA$ double mutant should be made and competed against the wildtype lacZ- parent strain. If the authors hypothesis is correct, then the CI for this competition should return to 1.

It would be valuable to present the total viable counts for each strain in the competitions shown in Fig. 6B-D in the main text or the supplement. The total growth in seawater should presumably be lower than when seawater is supplemented with glucose or fructose. But it is striking that the CI for seawater shows a sharp decline following 1 day of incubation, while a drop to the same CI take ~7 days when the medium is supplemented with fructose. Perhaps this is associated with some rescue of growth for the $\Delta fruT$ strain in the carbohydrate supplemented media? This comparison would be easier to make if the total viable counts for each strain were shown.

Fig. 6C-D are very nice data to indicate that FruT contributes to growth of Vc in the marine environment when fructose is present under aerobic conditions. The authors previous data indicates that aerobic conditions are required to prevent ArcA-dependent activation of FruI, which allows for FruT expression. To support this model, it would be compelling to demonstrate that the competitive effect observed in 6C is lost when cells are cultured under anaerobic conditions so that FruT expression is repressed.

It is quite striking that complemented strains throughout the study have phenotypes that perfectly mimic the WT, especially because these complementation constructs were expressed on multicopy vectors that likely would result in overexpression of the complemented genes. Was the native promoter used on these complementation vectors? This info should be added to the Methods. It is also striking that complementation in this manner allowed for recovery of WT CI in vivo. Because most plasmids are rapidly lost from *V. cholerae* in the absence of antibiotic selection. Was antibiotic included in vivo to maintain the plasmid? If so, did the competing strains have an empty vector control so that they were truly isogenic?

Responses to reviewers

Reviewer #1 (Remarks to the Author):

The paper exams the use of a PTS sugar by V. cholerae and specifically claim fructose uptake is essential for V. cholerae survival in seawater. Overall, the experiments are well done and appropriate controls were used, the results section is sound in their execution. However, the papers introduction and discussion are weak and the overall writing of the paper needs work.

The writing needs a lot of work.

Response:

Thank you for your patient and thoughtful reading as well as the constructive comments and advices about our manuscript. We have revised the manuscript based on your comments and suggestions. Especially, several parts in the Introduction and Discussion sections have been rewritten, including lines 53-54, 57-62, 106-107, 485-486, 489-493, 500-507, 509-517 and 519-534 of the revised MS.

Major comments

1. Spelling out V. cholerae throughout the paper, Vc not appropriate.

Response:

“Vc” has been changed to “*V. cholerae*” throughout the revised MS.

2. Line 140-142. I do not know how this data shows VCA0673 regulates fructose

Response:

In the revised MS, the content in lines 140-142 of the original MS **has been changed to** “BlastP analysis showed that VCA0673 shares sequence similarity to LacI-type regulators FruR, TreR, MalR and RbsR, which regulate the transport and/or utilization of fructose, trehalose, maltose and ribose, respectively” (lines 136-138 of the revised MS).

In addition, we have performed additional experiments to analyze whether mutation of VCA0673 affects the growth of *V. cholerae* in M9 medium supplemented with

33 each of those four sugars (fructose, trehalose, maltose and ribose) as the only carbon
source. The result showed that mutation of VCA0673 only influences the transport
and/or utilization of fructose, but not other three sugars (lines 142-150, Fig. 1a, and
Supplementary Fig. 1a-i of the revised MS).

**3. Line 147 fructose levels 30mM are very high in these experiments should be 10**
**mM.**

**Response:**

Some previous studies used sugar at a 0.5% (wt/vol) final concentration (about
27.8mM) in M9 medium (*Proc Natl Acad Sci U S A.* 2018. 115(21): E4796-E4805;
*Appl Environ Microbiol.* 1998. 64(10): 3784–3790; *Proc Natl Acad Sci U S A.* 2007
Mar 6; 104(10): 4124–4129). In addition, M9 Medium with 0.5% sugar was used
previously for analyzing the influence of sugars on the growth of *V. cholerae* (*Front*
*Microbiol.* 2018, 8:2703; *mBio.* 2020, 11(5): e01989-20; *J Bacteriol.* 2010,
192(12):3055-67). So, in the original MS, we used M9 medium supplemented with
30mM fructose to analyze whether deletion of *fruI* or *fruT* influences the growth of *V.*
*cholerae* with fructose as the only carbon source.

According to your suggestions, we have now performed additional experiments to
analyze the growth of $\Delta fruI$, $\Delta fruI^+$, $\Delta fruT$, $\Delta fruT^+$, Δfpr , ΔEI , $\Delta fruT\Delta fpr$ and WT in
M9 medium supplemented with 10mM fructose or other sugars (lines 142-150,
169-174, 186-193 and Supplementary Fig. 1a, c, e, g, i, j, m, o of the revised MS).
The results of these experiments were highly similar to that of experiments performed
in M9 medium supplemented with 30mM fructose or other sugars. We have now
added these results to the revised MS (lines 142-150, 169-174 and 186-193 of the
revised MS).

**4. Line 331 fructose concentration stated as 20mM**

**Response:**

When analyzing the influence of *fruI* and *fruT* on bacterial virulence, we used 20 mM
fructose to mimic the concentration of fructose in the small intestine environment,
which was mentioned once in the original MS (lines 238-239 of the original MS). We
have now added this information to several places of the revised MS (lines 242-243,

296 and 343 of the revised MS).

**5. Other sugars should be tested**

**Response:**

In addition to fructose, we have now performed additional experiments to analyze
whether mutation of *fruT* affects the growth of *V. cholerae* in M9 medium
supplemented with 30 mM, 10 mM, 3 μ M or 0.1 μ M of other sugars, including
glucose, trehalose, maltose and ribose (lines 409-412 and Supplementary Fig. 3a-p of
the revised MS). These data indicate that FruT does not contribute to the uptake of
these sugars.

**6. Referencing in general is very poor and many references are not appropriate or**
**incorrect. Many statement of fact are not referenced.**

**Response:**

We have now checked all references in the manuscript. Several references in the
original MS (Ref. 5-7, 17, 37-39 and 43 of the original MS) have been removed in the
revised MS, and several new references have been added to the revised MS (Ref. 5,
7-9, 11, 19-21, 39-43, 48-51, 53-56 and 59 of the revised MS). Please also see our
responses to your comments #7, #8, #9 and #11.

**7. Line 54. Worden AZ not an appropriate reference for this statement**

**Response:**

There were two references (Ref.4 and Ref.5) for the statement in line 54 of the
original MS. You are right, Ref.5 of the original MS is not appropriate, and has been
deleted in the revised MS.

**8. Line 55. Faruque SM discuss virulence genes among environmental strain, this**
**paper is a review and does not examine persistence**

**Response:**

The original statement “Long-term survival and persistence in the environment play
an important role in the *V. cholerae* lifecycle” (lines 55 of the original MS) **has been**
**changed to** “*V. cholerae* exhibits the ability to survive in the aquatic environment
98 year-around” (lines 53-54 of the revised MS). Two new references have now been

used to support this statement (Ref. 5 and Ref. 6 of the revised MS).

*9. Line 58 Reference not suitable, this paper measure DOC from a single water*
*sample from the Netherlands, not representative of seawater.*

**Response: :**

Ref.7 of the original MS **has been replaced** by Ref. 7 of the revised MS (lines 56-57
of the revised MS), which reported DOC (dissolved organic carbon) concentration of
49 individual seawater samples collected by a number of cruises in Pacific Ocean.

*10. Line 61 this statement is not true, many studies have examined nutrient uptake*
*by V. cholerae and similarly lines 433-437.*

**Response:**

Thank you for your comments. You are right, some previous studies showed that
extracellular DNA and dissolved organic matter of seawater may also support the
growth of *V. cholerae*. We have now added this information to the revised MS.

The original statement “Although *V. cholerae* employs some strategies for acquisition
and storage of nutrients in this environment, such as utilizing chitin in the exoskeleton
of marine crustaceans and using accumulated intracellular glycogen stores, it remains
largely unknown how pandemic *V. cholerae* efficiently uptakes and utilizes carbon
sources in seawater” (lines 57-62 of the original MS) **has been changed to** “Several
strategies enabling *V. cholerae* to acquire carbon sources in this environment have
been reported, such as utilizing chitin in the exoskeleton of marine crustaceans and
using accumulated intracellular glycogen stores. In addition, extracellular DNA and
dissolved organic matter of seawater may also support the growth of *V. cholerae*.

However, the mechanisms that *V. cholerae* utilizes to efficiently uptakes and
metabolizes carbon sources of seawater are still not fully understood” (lines 57-62 of
the revised MS).

The second part of the sentence in lines 433-437 of the original MS (“and it remains
largely unknown how pandemic Vc efficiently uptakes and utilizes carbon sources in
seawater”) **has been deleted** in the revised MS.

*11. Line 66-69 needs a reference*

**Response:**
Done (line 69 and Ref. 11 of the revised MS).

*12. Line 209-110. I could not find this reference and since the paper hinges on the*
*assumption that fructose is limiting in seawater as opposed to other sugars. Also is*
*this the only paper to show this from 1980?*

**Response:**
We are sorry for the confusion of the relevant statement in the original MS. In fact,
fructose and glucose are two major sugars in seawater, whose concentration
(0.12-0.55 μM and 0.03-0.67 μM , respectively) is much higher than that of other
sugars (0.01-0.24 μM) in seawater. This information was mentioned in the original
MS (lines 486-488 of the original MS).

To make it clear, we have now rewritten the relevant sentence. “Fructose of low
concentration is a major carbon source in seawater” (lines 109-110 of the original MS)
**has been changed to** “Fructose, whose concentration is in the range of 0.12-0.55 μM ,
is a major sugar present in seawater” (lines 106-107 of the revised MS).

Ref. 24 of the original MS (Ref. 26 of the revised MS) analyzed the concentration of 9
sugars in 11 seawater samples from different areas. It is the only paper that we found
to report the composition and concentration of different sugars in seawater.

*13. Line 440-445. Not sure how this relates to the present study and the data*
*described*

**Response:**
We agree with your comments. The content of lines 440-445 of the original MS has
been removed in the revised MS.

In addition, we have now combined following paragraph with the last paragraph of
the original MS (lines 433-439 and 486-490 of the original MS) into one (lines
462-470 of the revised MS), after deletion of lines 440-445 of the original MS.

14. Line 460 what is GI-7, this is the first mention. Also the whole paragraph is out

*of place. No mention of VPI-1 or VPI-2 and what they encode and how they related*
*to virulence and metabolism*

**Response:**

Thank you for your comments. We have now rewritten this paragraph.

i) In the original MS, GI-7 was first mentioned in the Result section (line 165 of the
original MS). To make it clear, “*fruT* and *fruI* are part of GI-7” (line 460 of the
original MS) **has been changed to** “Both *fruT* and *fruI* are located within a predicted
genomic island (GI-7)” (line 484 of the revised MS).

ii) In this paragraph of the original MS, we discussed GIs of *V. cholerae* in general,
but did not provide information about virulence genes in these GIs. Now we have
added these information here (lines 489-493 of the revised MS).

**15. Line 472 No mention of previously described work on CRP and cAMP or PTS**
**sugars in general in *Vc*. Significant work has been performed by others?**

**Response:**

Thank you for your suggestion. We have now added description for previous
significant work about cAMP-CRP and PTS in *V. cholerae* to the Discussion section
(lines 500-512 of the revised MS).

**16. Line 24 reference should ready available.**

**Response:**

We enclosed the copy of Ref. 24 of the original MS (Ref. 26 of the revised MS) as an
attachment of this letter.

**Reviewer #2 (Remarks to the Author):**

*In the manuscript by Liu et al, the authors characterize a novel fructose utilization*
*locus in pandemic clones of *V. cholerae* composed of a repressor, FruI, and*
*transporter, FruT. The authors demonstrate that FruT aids growth of *V. cholerae* in*
*the marine environment, which is perhaps the most convincing data in the*
*manuscript. They also show that the expression of FruT during infection may*
*marginally reduce the fitness of *V. cholerae* in vivo. They show that the presence of*

*FruT slightly alters virulence gene expression and attempt to mechanistically tie*
*this observation to modulation of cAMP levels, however, the data presented do not*
*provide a strong link between these phenotypes. Furthermore, the authors identify*
*that the O₂-sensing regulator ArcA activates expression of FruI under anaerobic*
*conditions to prevent FruT expression. The data to link ArcA-dependent regulation*
*of FruT to the phenotypes observed in vivo, however, are not convincing. While the*
*study is clearly written, there are a number of key controls or comparisons missing*
*that make it difficult to validate the model presented. If these are addressed, I*
*believe this manuscript could make a valuable addition to the field. Specific*
*comments, questions, and suggestions for the authors to consider are listed below.*

**Response:**

Thank you for your patient and thoughtful reading as well as the constructive
comments and advices about our manuscript. We have revised the manuscript based
on your comments and suggestions.

**Re:** “*They show that the presence of FruT slightly alters virulence gene*
*expression and attempt to mechanistically tie this observation to modulation of*
*cAMP levels, however, the data presented do not provide a strong link between these*
*phenotypes*”. To provide more evidence to show that FruT influences virulence gene
expression via modulating intracellular cAMP level, we have now performed several
additional experiments according to your suggestion in comment #9. i) In the original
MS, we showed that $\Delta fruT$ exhibited significantly lower intracellular cAMP level than
WT and the expression of virulence genes were upregulated in $\Delta fruT$ compared to WT.
Now we have performed additional experiment to show that the upregulation of
virulence gene expression in $\Delta fruT$ was inhibited by addition of 1mM extracellular
cAMP (this concentration has been used to analyze the influence of cAMP on
bacterial phenotypes (*Mol Microbiol.* 1996. 21(5):941-52; *Appl Environ Microbiol.*
2012. 78(2):411-9)) to the culture medium (lines 304-306 and Fig. 3p of revised MS).
These results indicate that FruT influences virulence gene expression via regulating
the intracellular cAMP level. ii) We have now constructed a double mutant $\Delta cya\Delta fruT$
and performed additional experiments to show that there was no difference in the
expression of virulence genes among Δcya , $\Delta cya\Delta fruI$ and $\Delta cya\Delta fruT$ in either the
small intestine of mice or M9 medium supplemented with 20 mM fructose (lines
318-320 and Supplementary Fig. 2i-l of revised MS). This suggests that deletion of

*fruI* and *fruT* in Δ *cya* background did not influence the expression of virulence genes,
indicating both FruT and FruI influence virulence gene expression via modulating the
intracellular cAMP level. iii) We also have performed additional experiment to show
that Δ *fruT* Δ *cyaA* competed evenly with Δ *cyaA* in the small intestine of mice (lines
314-316 and Supplementary Fig. 2h of the revised MS). This is in consistent with the
result that Δ *fruI* Δ *cyaA* competed evenly with Δ *cyaA* in the small intestine of mice of
the original MS, and further confirms that the influence of FruT and FruI on virulence
is dependent of their regulation on the intracellular cAMP level. Please also see the
response to your comment #9.

**Re: “The data to link ArcA-dependent regulation of FruT to the phenotypes**
**observed in vivo, however, are not convincing”**. ArcA is a global transcription
regulator in the *Enterobacteriaceae*, which controls the expression of genes involved
in several different pathways. For instance, ArcA directly regulates the expression of
*vpsT* involved in biofilm formation of *V. cholerae* which contributes to bacterial
intestinal colonization. It is highly likely that the influence of ArcA on the virulence of
*V. cholerae* might only be partially dependent of its regulation on *fruI* and *fruT*
expression. According to your suggestion in comment #12, we have now constructed
a double mutant Δ *arcA* Δ *fruT* and performed competitive infection assays in mice
between WT and Δ *arcA* Δ *fruT*. The results showed that the CI value of the double
mutant Δ *arcA* Δ *fruT* versus WT was 0.41 in the small intestine of mice, which was
significantly higher than the CI value (0.16) of Δ *arcA* versus WT, and significantly
lower than the CI value (3.18) of Δ *fruT* versus WT. These data confirm that in
addition to *fruI* and *fruT*, ArcA also influences the virulence of *V. cholerae* via other
regulatory pathways. These results have now been presented in the revised MS (lines
377-387 of the revised MS). Please also see the response to your comment #12.

**Re: “there are a number of key controls or comparisons missing that make it**
**difficult to validate the model presented”**. According to your suggestion in comment
#8, we have now performed additional experiments to use bacteria growing in M9

medium supplemented with 20 mM glucose as a negative control in several assays,
including qRT-PCR analysis on the expression levels of virulence genes in WT, $\Delta fruT$
and $\Delta fruI$ (lines 253-257 and Supplementary Fig. 2a of the revised MS), western
blotting analysis on the production of cholera toxin in WT, $\Delta fruT$ and $\Delta fruI$ (lines
253-257 and Supplementary Fig. 2b of the revised MS), qRT-PCR analysis on the
expression level of *fpr* in WT, $\Delta fruI$ and $\Delta fruT$ (lines 277-279 and Supplementary Fig.
2c of the revised MS), western blotting analysis on the Fpr protein level in WT, $\Delta fruI$
and $\Delta fruT$ (lines 277-279 and Supplementary Fig. 2d of the revised MS), analysis on
the phosphorylation status of Fpr in WT, $\Delta fruI$ and $\Delta fruT$ (lines 283-285 and
Supplementary Fig. 2e of the revised MS), and analysis on the cAMP concentration in
WT, $\Delta fruI$ and $\Delta fruT$ (lines 298-300 and Supplementary Fig. 2f of the revised MS).
The results showed that in contrast to the situation in M9 medium supplemented with
20 mM fructose, deletion of *fruT* or *fruI* has no significant effect on the virulence
gene expression, cholera toxin production, *fpr* expression, phosphorylation status of
Fpr and the intracellular cAMP level in M9 medium supplemented with 20 mM
glucose. Please also see the response to your comment #8.

*1. Fig. 1C – these differences are very subtle. Unless they are statistically significant*
*it seems inappropriate to comment on them. Or at the very least, statements about*
*these differences in the text must be qualified to make it clear that they are not*
*statistically significant.*

**Response:**

We are sorry that we did not perform statistical analyses for the data shown in Fig. 1a
and 1c of the original MS. Now we have performed statistical analyses, using the
original data in these figures, by two-way ANOVA. The difference between growth
curves of WT and $\Delta fruT$ and between growth curves of WT and $\Delta fruI$ are statistically
significant, while the difference between growth curves of WT and $\Delta fruT^+$ and
between growth curves of WT and $\Delta fruI^+$ are not statistically significant. The results
of statistical analyses have now been shown in Fig. 1a and Fig. 1c of the revised MS.

*2. Previous studies have shown that deletion of Fpr and/or EI prevents*
*growth/fermentation of El Tor V cholerae on fructose (ref 14 and 15). This contrasts*
*with the results obtained in Fig. 1D, which show that a Δfpr strain can still grow on*
*fructose. Can the authors comment on this discrepancy? Does an EI mutant exhibit*

*a similar degree of growth on fructose as the Fpr mutant?*

**Response:**

Thank you for your comments.

For Fpr, Ref. 15 of the original MS (Ref. 17 of the revised MS) showed that Fpr is
involved in the fructose transport in *V. cholerae* using an MM agar plates containing a
pH indicator and supplemented with fructose by looking at color change (medium
acidification upon sugar fermentation leads to a yellow color). The authors mentioned,
in Ref.15 of the original MS, that "*the colonies formed by ΔFPr mutants eventually*
*became yellow on indicator plates containing fructose*" (lines 4-7 on the right side of
page 1486 in Ref. 15), which indicates that Δfpr still can utilize fructose. This is in
consistent with our result.

For EI, Ref. 15 of the original MS (Ref. 17 of the revised MS) showed that ΔEI is
involved in the fructose transport by the agar plate-based fructose fermentation assay,
but not mentioned whether ΔEI can utilize fructose. Ref.14 of the original MS (Ref.
16 of the revised MS) analyzed bacterial growth curves in M9 medium supplemented
with 1% (55.6 mM) fructose in 800 mins, which showed that mutation of EI inhibited
the growth of *V. cholerae*, while Fig. 1E of Ref. 14 showed that OD₆₀₀ value of ΔEI
still exhibited a feeble increase from about 0.08 to 0.14 in 800 mins.

We have now constructed a ΔEI mutant and performed additional experiments to
confirm the growth of ΔEI in M9 medium supplemented with 30mM or 10mM
fructose in 1500 mins. The OD₆₀₀ value of ΔEI increased from 0.087 to 0.103 or 0.102
in 800 mins in M9 medium supplemented with 30mM or 10mM fructose, which is
similar to the result shown in Ref.14 of the original MS, indicating ΔEI can utilize
fructose. Furthermore, our results showed that the OD₆₀₀ value of ΔEI finally reached
0.154 or 0.134 in 1500 mins in those two conditions (lines 186-189 and
Supplementary Fig 1m, n of the revised MS), confirming ΔEI is able to utilize
fructose.

In addition, our results showed that ΔEI exhibited a slower growth rate than Δfpr in
M9 medium supplemented with 30mM or 10mM fructose. It is consistent with that
EI is required for the transport of all PTS substrates, and other PTSs in *V. cholerae*
may also participate in the fructose transport (*Nucleic Acids Res.* 2021, 49(3):
1397-1410).

*3. Fig. 2 – are these differences statistically significant? Error bars seem to overlap*
*between samples in this experiment, although it is unclear what these error bars*
*signify. This information should be added to the figure legend. Statistical*
*comparisons should be performed and commented on in the text.*

**Response:**

We are sorry that we did not perform statistical analyses for the data shown in Fig. 2
of the original MS. We have now performed statistical analyses for different
extracellular pH curves by using two-way ANOVA in this Fig. The difference
between extracellular pH curves of WT and $\Delta fruT$ are statistically significant, while
the difference between extracellular pH curves of WT and $\Delta fruT^+$ are not statistically
significant. The results of statistical analyses have now been shown in Fig.2 of the
revised MS, and mentioned in the text (line 212 of the revised MS).

Error bars represent mean \pm SD (n = 3), and this information has now been added to
the Legend for Fig.2 of the revised MS. In addition, the raw data for this Fig has now
been added to Source Data file of the revised MS.

*4. Why wasn't the $\Delta fruI \Delta fruT$ mutant competed against the parent strain in Fig.*
*3C? I believe that the expected outcome based on the authors model would be a*
*mutant that phenocopies the $\Delta fruT$ mutant. The comparison made in Fig. 3C is fine,*
*however, it requires an additional and unnecessary logical step by competing this*
*strain against a $\Delta FruT$ parent. Which leaves me wondering whether competition*
*between the lacZ- parent and the $\Delta fruI \Delta fruT$ strain did not show the expected*
*phenotype.*

**Response:**

In the original MS, the purpose of Fig. 3C was to investigate whether FruI contributes
to the intestinal colonization of *V. cholerae* via FruT, by doing competitive infection
assay between $\Delta fruI\Delta fruT$ and $\Delta fruT$. Some previous studies have used competitive
infection assays between double mutant and single mutant for investigating whether
two genes have additive effect on virulence or influence the virulence via a single
pathway (*EMBO J.* 2000, 19(13):3235-49; *Cell Host Microbe.* 2013, 14(6): 652–663).
Fig.3C of the original MS showed that $\Delta fruI\Delta fruT$ and $\Delta fruT$ competed evenly *in vivo*,
indicating *fruI* regulates the virulence of *V. cholerae* via FruT.

You are right, comparisons between the CI value of double mutant versus WT and the
CI value of single mutant versus WT, might be a better way to study this. We have
now performed an additional competitive infection assay between $\Delta fruI\Delta fruT$ and WT
in the small intestine of mice. As expected, the CI value of $\Delta fruI\Delta fruT$ versus WT was
3.26, which is similar to the CI value of $\Delta fruT$ versus WT (CI=3.18). It is in consistent
with the fact that $\Delta fruI\Delta fruT$ competed evenly with $\Delta fruT$ in the small intestine of
mice (shown in Fig. 3E of the original MS), and further confirms that FruI contributes
to the intestinal colonization of *V. cholerae* via FruT. These results have now been
presented in the revised MS (lines 229-232 and Fig. 3c, e of the revised MS).

*5. Fig. 3E – relative differences in transcript abundance are subtle for all of the*
*assays in Fig 3. Also, the in vitro growth conditions used in 3E would not be*
*expected to induce the expression of virulence genes in an El Tor V cholerae strain.*
*El Tor strains require growth in AKI medium to successfully activate the virulence*
*regulatory program. As a result, these data should be further supported via Western*
*blot analysis to show that the expression of key virulence proteins are altered to*
*mirror the changes observed in transcript abundance. Detection of cholera toxin or*
*TcpA is commonly used in the field.*

**Response:**

Thank you for your comments. We agree with that *in vitro* growth conditions (M9
medium supplemented with 20 mM fructose) used in experiments for Fig. 3E of the
original MS would not be expected to induce the expression of virulence genes. This

may result in the influence of *fruT* and *fruI* mutation on the expression of virulence
genes *in vitro* shown in Fig.3E of the original MS was subtle (However, these
differences are statistically significant).

AKI medium benefits the expression of virulence gene in *V. cholerae*. However,
different from M9 medium, AKI medium contains yeast extract, which may contain
fructose. Thus, AKI medium is not appropriate as culture medium for analyzing the
influence of *fruT* and *fruI* on virulence gene expression.

According to your suggestion, we have now performed additional western blotting
assays to analyze the production of cholera toxin, a major virulence factor of *V.*
*cholerae*, in WT, $\Delta fruI$ and $\Delta fruT$ in M9 medium supplemented with 20 mM fructose
or 20 mM glucose. The results showed that the production of cholera toxin was
significantly higher or lower in $\Delta fruT$ or $\Delta fruI$ compared to WT in M9 medium
supplemented with 20 mM fructose, while there was no difference in the production
of cholera toxin among WT, $\Delta fruT$ and $\Delta fruI$ in M9 medium supplemented with 20
mM glucose (lines 245-247, 253-257 and Fig. 3i and Supplementary Fig. 2b of the
revised MS). These results further confirm that FruT and FruI influence expression of
virulence factors via modulating bacterial fructose uptake.

***6. Fig. 3H and 3I. Transcript abundance of Fpr is supposed to be elevated ~2-fold***
***in $\Delta fruT$ according to 3H, however, the protein abundance for Fpr does not***
***increase 2-fold in 3I. If anything, this blot would indicate that protein abundance***
***for Fpr might actually be decreased in $\Delta fruT$. Although this comparison is difficult***
***to make because a loading control is not included. Regardless, the relevance of the***
***transcript abundance changes observed are unclear. Based on the increase in***
***unphosphorylated Fpr observed in the $\Delta fruT$ mutant, it appears as though FruT***
***does have some regulatory effect of fructose transport through the PTS. This is***
***assuming that the relative changes in the abundance of unphosphorylated Fpr to***
***phosphorylated Fpr observed are reproducible and significantly different based on***
***quantification of band intensities, which has not been performed. But FruT***
***influencing PTS transport through regulation of Fpr expression does not seem to be***
***the underlying mechanism.***

**Response:**

Thank you for your comments. The purpose of Fig. 3I of the original MS was to
analyze the phosphorylation status of Fpr in each single sample by comparing the
level of phosphorylated and non-phosphorylated Fpr in that sample, but not to
compare Fpr protein level among different samples. So, there was no loading control
used in that analysis, and total protein quantitation was also not performed.

According to your suggestion, we have now performed additional western blotting
assays to analyze the protein level of Fpr in WT, $\Delta fruT$ and $\Delta fruI$. RNAP (RNA
polymerase) was used as the loading control in this experiment. The result showed
that Fpr protein level exhibited a significant increase in $\Delta fruT$, but exhibited a
significant decrease in $\Delta fruI$ compared to that of WT in M9 medium supplemented
with 20 mM fructose, while Fpr protein level in $\Delta fruT$ or $\Delta fruI$ exhibited no difference
compared to that of WT in M9 medium supplemented with 20 mM glucose (lines
274-279, Fig. 3I and Supplementary Fig. 2d of the revised MS). This result is in
consistent with that shown in Fig.3H of the original MS, which compared the *fpr*
expression among $\Delta fruT$, $\Delta fruI$ and WT by qRT-PCR assays.

The western blotting assays for analyzing the phosphorylation level of Fpr shown in
the original MS were independently performed three times. According to your
suggestions, we have now performed additional analyses to determine the ratio of
phosphorylated Fpr to unphosphorylated Fpr by analyzing the band intensities. The
result showed that the phosphorylation level of Fpr in $\Delta fruT$ and $\Delta fruI$ growing in M9
medium supplemented with 20 mM fructose was significantly decreased or increased
compared to that of WT, while Fpr of $\Delta fruT$, $\Delta fruI$ and WT exhibited no
phosphorylation in M9 medium supplemented with 20 mM glucose. These results
have now been added to the revised MS (lines 283-285, Fig. 3m and Supplementary
Fig. 2e of the revised MS).

It is highly likely that there is a compensatory relationship between FruT and fructose
PTS. The repression of either of these two fructose transports leads to the increased
expression of the other transporter. We hypothesize that the reciprocal regulation
between fructose PTS and FruT maybe mediated via the change of bacterial
intracellular fructose concentration, which will be the subject of future studies. This
information has now been added to the revised MS (lines 528-534 of the revised MS).

Please also see the response to your comment #7.

*7. The authors demonstrate that FruI and FruT may impact PTS transport of*
*fructose through deletion of those factors. But does the PTS influence FruI/FruT*
*expression or activity?*

**Response:**

To analyze whether fructose PTS influences the expression of *fruT* and/or *fruI*, we
analyzed the expression of *fruT* and *fruI* in Δfpr and WT in M9 medium supplemented
with 20 mM fructose or 20 mM glucose. The results showed that the deletion of *fpr*
led to the elevated expression of *fruT*, but not *fruI*, in M9 medium supplemented with
20 mM fructose, while the expression of both *fruT* and *fruI* in Δfpr exhibited no
significant change compared to that of WT in M9 medium supplemented with 20 mM
glucose. These data indicate that fructose PTS also influences the expression of *fruT*
but not *fruI* via modulating bacterial fructose uptake (lines 523-528 and
Supplementary Fig. 6a, b of the revised MS).

Compensatory relationships have been observed between glucose transporters in
mammalian cells, however, the underlying mechanisms are unclear. It is highly likely
that there is a compensatory relationship between FruT and fructose PTS. The
repression of either of these two fructose transports leads to the increased expression
of the other transporter. The exact compensatory mechanism between FruT and
fructose PTS is currently unclear, however, our data showed that FruI does not play a
role in that mechanism. We hypothesize that the reciprocal regulation between
fructose PTS and FruT maybe mediated via the change of bacterial intracellular
fructose concentration, which will be the subject of future studies. These contents
have now been added to the Discussion section of the revised MS (lines 528-534 of
the revised MS).

*8. Throughout the study, mutant strains are largely studied in minimal medium with*
*fructose as a sole carbon source. A relevant negative control condition would be*
*beneficial in many instances to demonstrate that the phenotypes observed are*
*specific to growth on fructose. Growth in minimal medium with glucose (PTS*
*transported) or maltose (ABC transported) could be relevant negative controls. For*
*example, in Fig. 3I it would be valuable to examine samples grown in something*

*other than fructose to ensure that the non-phosphorylated Fpr observed is a direct*
*result of fructose uptake through the PTS as hypothesized. Similarly, transcript*
*abundance of virulence genes for strains grown in glucose would help support the*
*conclusion that differences observed are specific to fructose uptake by FruT as*
*hypothesized. A control like this is actually included in Fig. 6D, and is highly*
*valuable in the context of that experiment.*

**Response:**

We totally agree with your suggestions.

We have now performed additional experiments to use bacteria growing in M9
medium supplemented with 20 mM glucose as a negative control in several other
assays, including qRT-PCR analysis on the expression level of virulence genes in WT,
$\Delta fruT$ and $\Delta fruI$ (line 256 and Supplementary Fig. 2a of the revised MS), western
blotting analysis on the production of cholera toxin in WT, $\Delta fruT$ and $\Delta fruI$ (line 256
and Supplementary Fig. 2b of the revised MS), qRT-PCR analysis on the expression
level of *fpr* in WT, $\Delta fruI$ and $\Delta fruT$ (line 279 and Supplementary Fig. 2c of the
revised MS), western blotting analysis on the Fpr protein level in WT, $\Delta fruI$ and
$\Delta fruT$ (lines 277-279 and Supplementary Fig. 2d of the revised MS), analysis on the
phosphorylation status of Fpr in WT, $\Delta fruI$ and $\Delta fruT$ (line 285 and Supplementary
Fig. 2e of the revised MS), and analysis on the cAMP concentration in WT, $\Delta fruI$ and
$\Delta fruT$ (line 300 and Supplementary Fig. 2f of the revised MS).

The results showed that in contrast to the situation in M9 medium supplemented with
20 mM fructose, deletion of *fruT* or *fruI* has no significant effect on the virulence
gene expression, cholera toxin production, *fpr* expression, phosphorylation status of
Fpr and the intracellular cAMP level in M9 medium supplemented with 20 mM
glucose. These results confirm the phenotypes observed are specific to growth on
fructose.

***9. The link between cAMP and virulence gene regulation with regard to fruT and***
***fruI is largely correlative and/or based on epistasis between fruI and cyaA in vivo***

(Fig. S2), the latter of which could be influenced by pleiotropic effects of the *cyaA*
deletion *in vivo*. If the effect of *fruT* on virulence gene expression occurs via altered
cAMP levels as the authors hypothesize, then addition of extracellular cAMP at
physiological levels should reduce the enhanced virulence gene expression observed
in the $\Delta fruT$ mutant. Also, deleting *cyaA* (or CRP) in the parent, $\Delta fruI$, and $\Delta fruT$
backgrounds (to make double mutants) should ablate differences in virulence gene
expression *in vitro* under conditions similar to those tested in Fig. 3E.

**Response:**

To provide more evidence that FruT influences virulence gene expression via
modulating the intracellular cAMP level, we have now performed several additional
experiments according your suggestion.

i) In the original MS, we showed that $\Delta fruT$ exhibited significantly lower
intracellular cAMP levels than WT and the expression of virulence genes were
upregulated in $\Delta fruT$ compared to WT in M9 medium supplemented with 20mM
fructose. Now we have performed additional experiment to show that the upregulation
of virulence gene expression in $\Delta fruT$ was inhibited by addition of 1mM extracellular
cAMP (concentration used to analyze the influence of cAMP on bacterial phenotypes
in other works (*Mol Microbiol.* 1996. 21(5):941-52; *Appl Environ Microbiol.* 2012.
78(2):411-9)) to the culture medium (lines 304-306 and Fig. 3p of the revised MS).
These results indicate that FruT influences virulence gene expression via regulating
the intracellular cAMP level.

ii) We have now constructed a double mutant $\Delta cya\Delta fruT$ and performed additional
experiments to show that there was no difference in the expression of virulence genes
among Δcya , $\Delta cya\Delta fruI$ and $\Delta cya\Delta fruT$ in either the small intestine of mice or M9
medium supplemented with 20 mM fructose (lines 318-320 and Supplementary Fig
2i-1 of revised MS). This suggest that deletion of *fruI* and *fruT* in Δcya background
did not influence the expression of virulence genes, indicating both FruT and FruI
influence virulence gene expression via modulating the intracellular cAMP level.

iii) We also have performed additional experiment to show that $\Delta fruT\Delta cyaA$
competed evenly with $\Delta cyaA$ in the small intestine of mice (lines 314-316 and
Supplementary Fig. 2h of the revised MS). This is in consistent with the result that
$\Delta fruI\Delta cyaA$ competed evenly with $\Delta cyaA$ in the small intestine of mice of the original
MS, and further confirms that the influence of FruT and FruI on virulence is
dependent of their regulation on the intracellular cAMP level.

**10. Expression of FruT is relatively low in vitro when FruI is intact according to**
**Fig. 1. This may explain why the phenotypes to demonstrate that FruT supports**
**growth on low fructose concentrations in Fig. 5 are very subtle. FruT expression is**
**greatly increased in a $\Delta fruI$ mutant. So, to more convincingly demonstrate that**
**FruT is really a fructose transporter that can promote fructose uptake at low**
**concentrations it might be valuable to assess the fructose uptake and growth of a**
**FruI mutant in experiments that parallel those shown in Fig. 5A-E. Testing a $\Delta fruI$**
**mutant would also be valuable in Fig. 2 for the same reason. Or alternatively,**
**simply overexpressing FruT from an inducible promoter might be even better**
**because it would avoid the pleiotropic changes in gene expression that occur in the**
**$\Delta fruI$ mutant.**

**Response:**

Thank you for your comments. According to your suggestions, now we have
constructed a *fruT*-overexpressing strain (*fruT*⁺⁺) by introducing a high-copy plasmid
containing *fruT* with Trc promoter into WT. When comparing the growth of *fruT*⁺⁺,
WT, $\Delta fruT$ and Δfpr , IPTG was added to a final concentration of 0.1mM. This
information has now been added to the Methods section of the revised MS (lines
557-559 and 581-582 of the revised MS).

The result showed that *fruT*⁺⁺ exhibited a growth advantage compared to WT in M9
medium supplemented with 0.1 μ M or 3 μ M fructose, and reached a higher final cell
counts than WT (lines 402-405 and Fig. 5a, b of the revised MS). In contrast, *fruT*⁺⁺
exhibited almost same growth ability as WT in M9 medium supplemented with 30
mM or 10mM fructose (lines 405-407 and Fig. 5c, d of the revised MS). In addition,
we showed that there was no significant difference in the growth among WT, $\Delta fruT$
and *fruT*⁺⁺ in M9 medium supplemented with 30 mM, 10 mM, 3 μ M or 0.1 μ M
glucose (lines 409-412 and Supplementary Fig. 3a-p of the revised MS). These results
further confirm that FruT contributes to fructose uptake at low fructose
concentrations.

**11. Fig. 5E – are these data representative of multiple experiments? Are the**
**differences statistically significant?**

**Response:**

Fig.5E of the original MS showed the data of three independent experiments.
Variations among the results of these three independent experiments were very small,
so error bars are not visible in this Fig. The raw data of this Fig has now been
presented in Source Data file of the revised MS.

We have now performed statistical analyses for data in this Fig using two-way
ANOVA. The difference between fructose utilization of WT and $\Delta fruT$ is statistically
significant, while the difference between fructose utilization of WT and Δfpr is not
statistically significant. The results of statistical analyses have now been shown in
Fig.5e of the revised MS.

**12. Deletion of ArcA may attenuate *V. cholerae* in vivo due to the depression of**
***FruT* as hypothesized. However, it is equally likely that ArcA may have a distinct**
**effect on *V. cholerae* virulence that is entirely independent of *FruT* expression. To**
**formally test this, a $\Delta fruT \Delta arcA$ double mutant should be made and competed**
**against the wildtype *lacZ*- parent strain. If the authors hypothesis is correct, then**
**the CI for this competition should return to 1.**

**Response:**

Thank you for your comments. You are right that ArcA is a global transcription
regulator in the *Enterobacteriaceae*, which controls the expression of genes
involved in several different pathways. For instance, ArcA directly regulates the
expression of *vpsT* involved in biofilm formation of *V. cholerae* which contributes to
bacterial intestinal colonization. It is highly likely that the influence of ArcA on the
virulence of *V. cholerae* might only be partially dependent of its regulation on *fruI* and
*fruT* expression. According to your suggestion, we have now constructed a double
mutant $\Delta arcA \Delta fruT$ and performed competitive infection assays in mice between WT
and $\Delta arcA \Delta fruT$. The results showed that the CI value of the double mutant
$\Delta arcA \Delta fruT$ versus WT was 0.41 in the small intestine of mice, which was
significantly higher than the CI value (0.16) of $\Delta arcA$ versus WT, and significantly
lower than the CI value (3.18) of $\Delta fruT$ versus WT. These data confirm that in
addition to *fruI* and *fruT*, ArcA also influences the virulence of *V. cholerae* via other

regulatory pathways. These results have now been presented in the revised MS (lines
377-387 of the revised MS).

*13. It would be valuable to present the total viable counts for each strain in the*
*competitions shown in Fig. 6B-D in the main text or the supplement. The total*
*growth in seawater should presumably be lower than when seawater is*
*supplemented with glucose or fructose. But it is striking that the CI for seawater*
*shows a sharp decline following 1 day of incubation, while a drop to the same CI*
*take ~7 days when the medium is supplemented with fructose. Perhaps this is*
*associated with some rescue of growth for the $\Delta fruT$ strain in the carbohydrate*
*supplemented media? This comparison would be easier to make if the total viable*
*counts for each strain were shown.*

**Response:**

Total viable counts for each strain in the competition assays shown in Fig. 6B-6D of
the original MS has now been presented in Source Data file of the revised MS. As you
expected, data of total viable counts showed that the survival of both $\Delta fruT$ and WT in
seawater was lower than that in artificial seawater medium supplemented with
fructose or glucose.

You are right that there were some rescue of growth for $\Delta fruT$ in artificial seawater
medium supplemented with fructose in the first three days of incubation, resulting in
that $\Delta fruT$ exhibited almost same survival ability as WT within that period. Although
the reason for the rescue of growth of $\Delta fruT$ in that condition remains unclear, it may
explain the slower decrease of CI value ($\Delta fruT$ versus WT) in artificial seawater
medium compared to that in seawater.

*14. Fig. 6C-D are very nice data to indicate that FruT contributes to growth of Vc in*
*the marine environment when fructose is present under aerobic conditions. The*
*authors previous data indicates that aerobic conditions are required to prevent*
*ArcA-dependent activation of FruI, which allows for FruT expression. To support*
*this model, it would be compelling to demonstrate that the competitive effect*
*observed in 6C is lost when cells are cultured under anaerobic conditions so that*
*FruT expression is repressed.*

**Response:**

Thank you for your comment. We have now performed additional experiments to
show that $\Delta fruT$ exhibited the same survival ability as WT when growing
anaerobically in artificial seawater medium supplemented with fructose (lines
446-448 and Fig.6e of the revised MS), indicating the competitive effect observed in
Fig.6C of the original MS is lost under anaerobic condition.

*15. It is quite striking that complemented strains throughout the study have*
*phenotypes that perfectly mimic the WT, especially because these complementation*
*constructs were expressed on multicopy vectors that likely would result in*
*overexpression of the complemented genes. Was the native promoter used on these*
*complementation vectors? This info should be added to the Methods. It is also*
*striking that complementation in this manner allowed for recovery of WT CI in vivo.*
*Because most plasmids are rapidly lost from V. cholerae in the absence of antibiotic*
*selection. Was antibiotic included in vivo to maintain the plasmid? If so, did the*
*competing strains have an empty vector control so that they were truly isogenic?*

**Response:**

Thank you for your comments. When constructing the complementation plasmid,
corresponding gene was cloned into a low-copy plasmid (pBAD33,10-15 copies per
cell) with its native promoter. The use of low-copy plasmids and native promoters to
construct complementation plasmid may have made the phenotypes of the
complemented strain mimic that of WT closely. This situation has been found in some
previous studies (*PLoS Pathog.* 2019. 15(8): e1007952; *Environ Microbiol.* 2020.
22(10):4231-4243; *BMC Microbiol.* 2011; 11: 72; *Int J Mol Sci.* 2019. 20(18): 4339).
The relevant information has been added to the Methods section of the revised MS
(lines 555 of the revised MS).

You are right, growth curves and extracellular pH curves of complemented strains
shown in this study (Fig. 1a, 1c, 2 and Supplementary Fig. 1i, 1j of the revised MS)
all perfectly mimic that of WT. It is worth noting that phenotypes of complemented
strains shown in other experiments of this study (Fig. 1b, 3d, 4e, 4g and
Supplementary Fig. 1l of the revised MS) are all slightly different from that of WT
(however, these differences were not statistically significant).

We did not use antibiotics and WT with an empty plasmid control in the animal
experiments shown in the original MS. You are right, plasmids maybe lost from *V.*
*cholerae in vivo* in the absence of antibiotic. According to your suggestions, we have
now re-performed the competition assays using WT with an empty plasmid pBAD33
(lines 631-633, Supplementary Tab. 2 of the revised MS). The CI value of $\Delta fruI+$
versus WT containing pBAD33 in the re-performed experiment was 1.19 (the
corresponding CI value is 1.13 in the original MS), and the CI value of $\Delta fruT+$ versus
WT containing pBAD33 in the re-performed experiment was 0.86 (the corresponding
CI value is 1.21 in the original MS). There is no significant difference between the CI
values obtained from re-performed and original assays. The results of competition
assays between WT and complemented strains in the original MS have now been
replaced by that of re-performed competition assays in the revised MS (lines 223-224,
227-228 and Fig. 3d of the revised MS).

REVIEWERS' COMMENTS

Reviewer #2 (Remarks to the Author):

The revised manuscript by Liu et al addresses all of the concerns I raised during the initial round of review. I especially want to commend the authors for the substantial additional experiments and controls performed in the revised manuscript. These allowed me to properly evaluate the data provided in the initial submission, and have helped to strengthen the conclusions that the authors make. I believe that this manuscript makes an important contribution to the field and is now well-suited for publication at Nature Communications.

I have only one minor comment for the authors to consider:
Fig. 3i and Fig. S2B – should be “cholera toxin” not “cholerae toxin”

Responses to reviewers

Reviewer #2 (Remarks to the Author):

The revised manuscript by Liu et al addresses all of the concerns I raised during the initial round of review. I especially want to commend the authors for the substantial additional experiments and controls performed in the revised manuscript. These allowed me to properly evaluate the data provided in the initial submission, and have helped to strengthen the conclusions that the authors make. I believe that this manuscript makes an important contribution to the field and is now well-suited for publication at Nature Communications.

I have only one minor comment for the authors to consider:

Fig. 3i and Fig. S2B – should be “cholera toxin” not “cholerae toxin”

Response:

Thank you for your consideration of this work. We appreciate your effort in helping us improve our manuscript.

“cholerae toxin” **has now been changed to** “cholera toxin” in Fig. 3i and Fig. S2B of the revised MS.